# Keep It in Mind: User-Centric Continual Spatial Intelligence Reasoning in Egocentric Video Streams

**Yun Wang** [1 2]  **Junbin Xiao** [* 2 3]  **Han Lyu** [4]  **Yifan Wang** [5]  **Jing Zuo** [6]
**Zhanjie Zhang** [7]  **Hong Huang** [1]  **Dapeng Wu** [1]  **Angela Yao** [3]

## Abstract

We introduce UCS-Bench, a dataset spanning 170+ hours of egocentric visual observations with 8.1K+ timestamped questions for diagnosing User-Centric Continual Spatial intelligence in egocentric video streams. UCS-Bench targets a new problem that emphasizes dynamic spatial reasoning, long-term memory, and their alignment with users' real-time locations. We propose DirectMe, a framework that incrementally constructs and maintains a structured spatial memory from streaming egocentric observations. DirectMe enables robust tracking and recall of object locations, all relative to the user's movement over time. By tightly coupling visual perception with memory updates and spatial reasoning, our approach supports long-horizon queries that require recalling interactions, resolving viewpoint-induced ambiguities, and adapting to dynamic scenes. Our experiments show that DirectMe significantly improves the spatial reasoning of leading multimodal LLMs; it also surpasses many spatially aware and long-form streaming video models. We hope our benchmark and solution will advance spatial intelligence research for egocentric AI assistants. Data and code are available at https://github.com/cocowy1/UCS-Bench.

## 1. Introduction

Humans routinely solve spatial problems that go far beyond instantaneous perception: as we walk through hallways, turn into rooms, step outside, and re-enter a building, we continuously keep track of *where we are* and how the past world environments are oriented relative to our body, even though they are out of sight (Plizzari et al., 2025). This everyday capability relies on *user-centric continual spatial intelligence*: the ability to *incrementally integrate streaming first-person observations* to maintain and update an egocentric spatial reference – our heading, orientation, and relative directions to places and objects - despite frequent viewpoint changes, motion blur, and occlusions. Building AI assistants that operate from wearable egocentric cameras would benefit enormously from user-centric spatial understanding, for example, they can answer "*The lift is behind you.*" for the question "*Where is the lift?*", for both robots (Anderson et al., 2018) and human users (especially those with visual impairments (Xiao et al., 2025)) when they initially faced the lift but then turned around, causing the lift to be out of view of the ego-camera at the query moment.

Despite its centrality to human cognition, user-centric continual spatial intelligence remains largely underexplored in modern CV and AI research. Most existing benchmarks and methods for visual spatial intelligence focus on images or offline short video clips, and the questions are from a third-person view for diagnosis purposes without sufficient in-situation engagement (Yang et al., 2025d; Zhang et al., 2025b; Yang et al., 2025b; Zheng et al., 2025; Wu et al., 2026; 2025c). In these cases, the spatial relations are largely limited to static indoor scenes, or to what is immediately visible that can be inferred using instantaneous visual cues, local geometry, or commonsense priors. More recent research has therefore progressed toward dynamic (Li et al., 2025c; Zhou et al., 2025a), more comprehensive (Lin et al., 2025b), and online spatial understanding (Yang et al., 2025e; Lin et al., 2025c) from continuous video streams. However, a key missing piece in these works is that they rarely address the problem of reasoning about the orientation and location of the environments (both indoor and outdoor, static and dynamic) relative to those of the users' real-time movements.

---

[1]Department of Computer Science, City University of Hong Kong [2]University of Science and Technology of China [3]National University of Singapore [4]The Chinese University of Hong Kong, Shenzhen [5]University of Electronic Science and Technology of China [6]Beijing University of Posts and Telecommunications [7]Zhejiang University. Correspondence to: Junbin Xiao <junbinxiao@ustc.edu.cn>.

*Proceedings of the 43rd International Conference on Machine Learning*, Seoul, South Korea. PMLR 306, 2026. Copyright 2026 by the author(s).

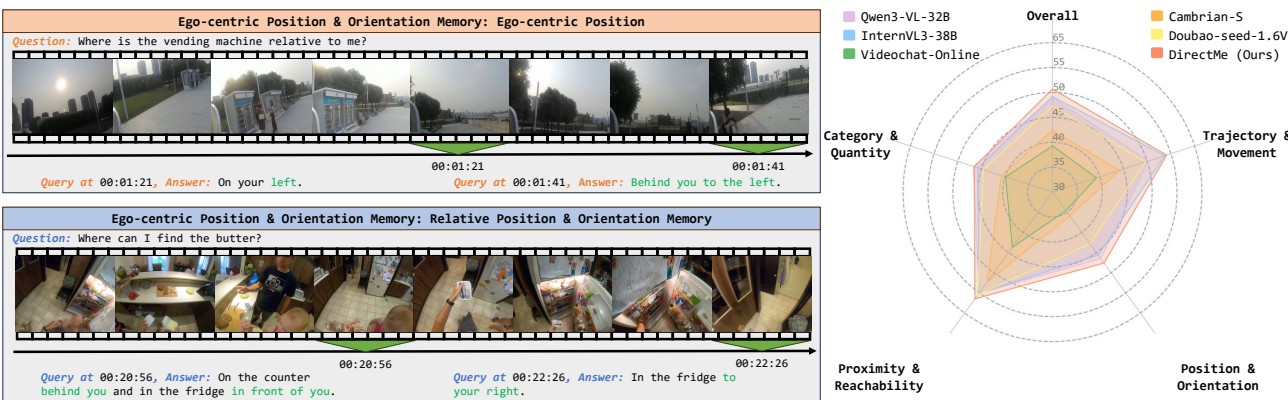

*Figure 1.* UCS-Bench focuses on user-centric continual spatial reasoning in egocentric video streams, requiring models to track evolving spatial relations relative to users' real-time locations. Existing models struggle to capture these fluid spatial evolutions, while our method DirectMe achieves significant improvements by maintaining an evolving and user-centric spatial scene graph memory.

To address this gap, we introduce **UCS-Bench**, an egocentric video question answering (QA) dataset that benchmarks U̲ser-centric C̲ontinual S̲patial intelligence reasoning for the first time. UCS-Bench contains 532 long videos spanning over 170 hours of egocentric visual observations encompassing both indoor and outdoor daily activities. We manually annotate about 8.1K QA pairs from 4 aspects of 1) *location & orientation*, 2) *path & movement*, 3) *distance & proportion*, and (4) *object & quantity recall*, all emphasizing user-centric continual spatial intelligence reasoning. The QA pairs are timestamp-specific and follow an online, streaming QA setting (Niu et al., 2025), mirroring how users would naturally interact with wearable AI assistants in practice. Examples are presented in Figures Figure 1 and 2.

By grounding questions in both past observations and the user's current spatial state, UCS-Bench exposes new challenges that are largely overlooked in existing episodic or clip-based spatial intelligence evaluations, and highlights the explicit mechanisms for maintaining and updating spatial memory. To this end, we propose **DirectMe**, a framework that incrementally builds and maintains a user-centric structured spatial memory from streaming egocentric observations. Rather than treating long videos as discrete frame sequences for MLLMs, DirectMe dynamically organizes spatial information (object locations, depths, and camera poses) following the user's real-time movement and maintains an evolving spatial scene graph as a cognitive map for MLLM reasoning, enabling consistent tracking of object locations and relations as the user moves around the environment. This design allows the model to recall spatial facts acquired long ago, reconcile them with new evidence, and answer long-horizon spatial queries without catastrophic forgetting.

Extensive experiments demonstrate that DirectMe substantially enhances the spatial reasoning capabilities of leading MLLMs (*e.g.*, Qwen3-VL (Bai et al., 2025a)). It also outperforms recent spatial-aware (*e.g.*, Cambrian-S (Yang et al., 2025e)) and long streaming video baselines (*e.g.*, Dispider (Qian et al., 2025)). UCS-Bench and DirectMe lay a foundation for studying user-centric continual spatial intelligence and pave the way for wearable egocentric AI assistants. Our major contributions are:

- We formulate user-centric continual spatial intelligence and contribute **UCS-Bench** as a benchmark for promoting related model capabilities.

- We propose **DirectMe**, a training-free framework that highlights the construction and maintenance of a dynamically evolving scene graph of the world as spatial memory to enhance streaming QA about continual spatial intelligence.

- We achieve new state-of-the-art on **UCS-Bench**, surpassing many well-established general-purpose and specialized MLLMs. We comprehensively analyze model results and provide new insights for future research.

## 2. Related Work

### 2.1. Benchmarking Spatial Intelligence

Existing spatial intelligence benchmarks can be broadly organized along two orthogonal axes: *input modality* and *protocol*, progressing from static, offline perception to temporal and online settings. Image-based benchmarks evaluate single-image spatial understanding or structured 3D reasoning (*e.g.*, SpatialRGPT-Bench (Cheng et al., 2024a), SpatialVLM (Chen et al., 2024), CV-Bench (Zhu et al., 2025b), BLINK (Fu et al., 2024), 3DSRBench (Ma et al., 2025)), while multi-image/multi-view benchmarks test cross-view consistency by reconciling partial observations offline (*e.g.*, MultiSPA (Xu et al., 2025), MMSI-Bench (Yang et al.,

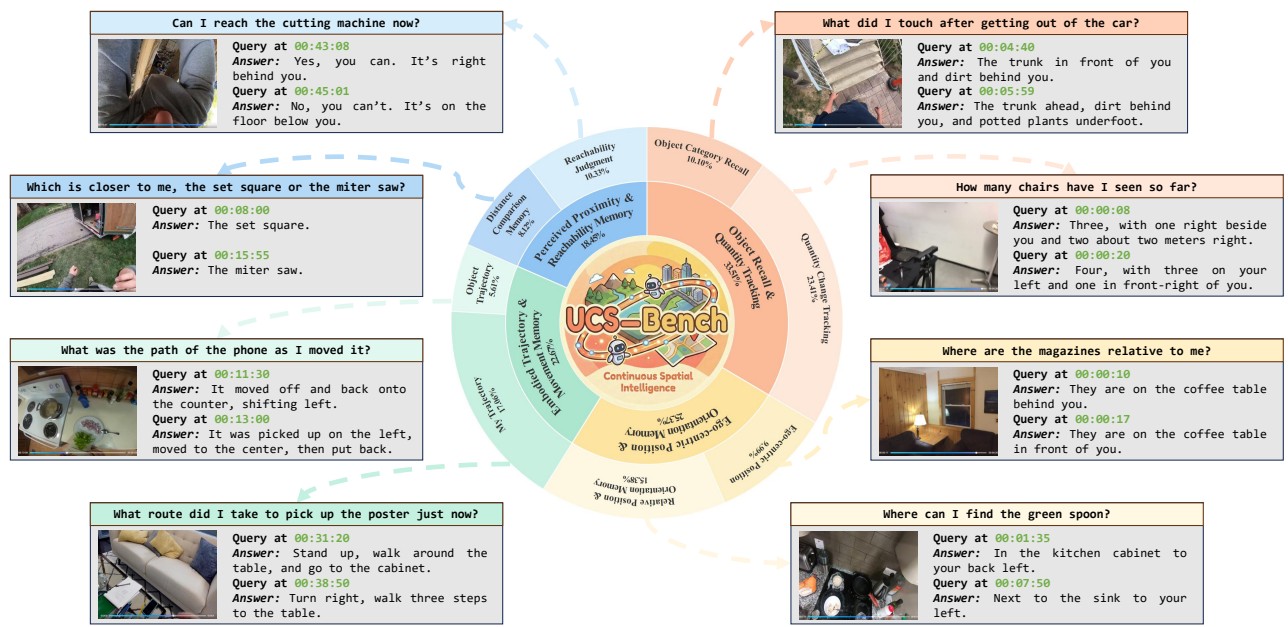

*Figure 2.* UCS-Bench features a comprehensive set of four major and eight minor categories of spatial cognitive tasks. Its hallmark is the evaluation of **continual spatial memory**, where the same query is posed at two different timestamps, requiring the model to dynamically update its spatial awareness and provide accurate, context-aware answers as the environment changes.

2025d), All-Angles Bench (Yeh et al., 2025), ViewSpatial-Bench (Li et al., 2025a), MindCube (Yin et al., 2025)). Video benchmarks introduce temporal cues to assess motion and object–camera/object–object relations, with some emphasizing egocentric or embodied spatial understanding (*e.g.*, VLM4D (Zhou et al., 2025b), VSI-Bench (Yang et al., 2025b), SPAR-Bench (Zhang et al., 2025b), STI-Bench (Li et al., 2025c), SpatialLadder (Li et al., 2025b), ).

Several efforts focus on online/streaming protocols under sequential observations. OST-Bench (Lin et al., 2025c) formulates streaming understanding as egocentric exploration with multi-turn QA that integrates the current view with accumulated context. VSI-Super (Yang et al., 2025e) targets arbitrarily long (and synthesized) videos and stresses long-horizon evidence accumulation for tasks such as object recall and counting. However, their core challenge is long-horizon recall and evidence accumulation, not explicitly tracking user-dependent spatial state under ego-motion (i.e., orientation-aware updates as the camera wearer moves and turns) as in our UCS-Bench.

### 2.2. Multimodal Large Language Models

Multimodal large language models (MLLMs) (Liu et al., 2024a; Bai et al., 2025a;b; Zhu et al., 2025a; Zhang et al., 2024; Wang et al., 2022) excel at short-horizon visual semantic understanding (Xiao et al., 2021), yet remain limited in coping with long streaming videos and spatial reasoning. To handle long streaming videos, recent advances highlight the

importance of memory modeling, either end-to-end learned (Zhang et al., 2025a; Huang et al., 2025) or training-free (Di et al., 2025; Kim et al., 2025; Xiao et al., 2026). However, their memory contents are limited to visual feature representations or KV-Cache, where spatial information, especially relative to the users' real-time locations, is largely overlooked.

To improve spatial reasoning in MLLMs, prior work can be organized by *where* spatial knowledge is injected. **(i) Data-centric supervision scaling** (Chen et al., 2024; Ray et al., 2024; Cheng et al., 2024a; Cai et al., 2025b; Yang et al., 2025e;c) constructs large-scale spatial instruction data for SFT by converting simulator state (e.g., 3D object poses and camera trajectories) and pseudo-spatial cues extracted from real images (e.g., depth, detections, segmentations) into template-driven spatial QA, or by aggregating and reorganizing open-source VQA datasets into spatially focused training corpora. **(ii) Geometry-augmented representations** (Wu et al., 2025a; Fan et al., 2025) instead modify the model's internal representation, introducing explicit geometric encoders, 3D tokens, or reconstruction-aligned features, which improves geometric reasoning but typically incurs additional computation and system complexity. **(iii) Strategy-augmented learning and reasoning** explores alternative ways to elicit spatial behaviors, including reinforcement learning (Ouyang et al., 2025), cognitive map-based mental modeling (Yin et al., 2025), tool-augmented (Cai et al., 2025a), or interleaved reasoning (Wu et al., 2025b), often targeting specific reasoning patterns.

*Table 1.* Video constitution of UCS-Bench.

| Dataset | Count | QA Pairs | Avg. Length (m) | Percentage (%) |
|---|---|---|---|---|
| Egolife | 189 | 2569 | 11.8 | 35.5 |
| HourVideo | 167 | 3087 | 42.8 | 31.4 |
| ScanNet | 60 | 612 | 0.3 | 11.3 |
| EPIC-KITCHENS | 56 | 1055 | 11.7 | 10.5 |
| EgoBlind | 38 | 588 | 2.2 | 7.1 |
| TeleEgo | 22 | 203 | 19.1 | 4.1 |
| **Total** | **532** | **8114** | **19.9** | **100.0** |

Despite this progress, existing spatial-centric MLLMs are still trained and evaluated in *static* or *episodic* settings. However, egocentric perception is inherently *streaming* and *user-centric*: models must continually update objects' spatial states relative to the camera wearer, even when they are out of sight. This gap motivates DirectMe, which underscores a structured memory module for robust continual spatial reasoning in egocentric video streams.

## 3. UCS-Bench

### 3.1. Benchmark Construction

We select 532 videos from six video sources: HourVideo (Chandrasegaran et al., 2024), ScanNet (Dai et al., 2017), EgoLife (Yang et al., 2025a), EgoBlind (Xiao et al., 2025), EPIC-KITCHENS (Damen et al., 2018) and TeleEgo (Yan et al., 2025). We select videos spanning short clips (seconds) to long recordings (hours), covering both indoor and outdoor scenes. Other details are presented in Table 1 and Appendix § F.1.

We follow a strict four-stage pipeline (see Figure 3): 1) stream preprocessing & unification → 2) taxonomy-driven QA authoring with evidence → 3) multiple-choice conversion → 4) quality control through bias diagnostics and human auditing. These stages are designed to ensure that questions are evidence-grounded while reflecting user-centric continual spatial intelligence.

**Stage 1: Stream preprocessing and unification.** We standardize videos from all sources into a unified metadata schema (*e.g.*, video_uid, duration, FPS, and timestamps) to enable data-source agnostic authoring and evaluation. Each question is anchored to a specific query timestamp. Annotators ensure that the correct answer requires integrating the spatial information from earlier observations rather than relying solely on the current frame.

**Stage 2: Taxonomy-driven QA authoring with evidence.** We require all questions to be centered on the users' orientations and locations relative to other things in the environments. Notably, the answer at a specific timestamp must recall earlier observations that may be out of sight and link them to the current observed states at the query moment.

We categorize user-centric continual spatial intelligence questions into four dimensions (see Figure 2), emphasizing

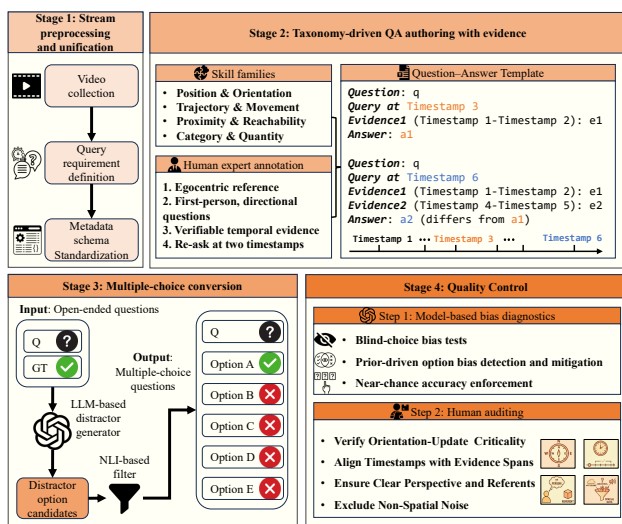

*Figure 3.* Overview of our benchmark construction pipeline.

egocentric reasoning under streaming observations and temporal updates: (**1**) **Position & Orientation:** Questions that track the user–object relative positions and facing directions from the user's viewpoint (e.g., left/right, front/behind), staying valid as the user turns or the scene evolves. (**2**) **Trajectory & Movement:** Questions that remember how the user or objects move over time, including paths, intermediate landmarks, and start/end locations across multiple moments. (**3**) **Proximity & Reachability:** Questions that assess distance and practical reachability from the user's current (or specified) viewpoint given the ongoing stream, accounting for obstacles and feasible navigation. (**4**) **Category & Quantity:** Questions that track item existence, categories, and counts from an everyday user perspective, updating as objects are moved, added, or removed.

We invite 50 trained university students to watch the videos and annotate QA pairs while adopting the camera wearer's perspective. Each answer is accompanied by one or more temporal segments serving as visual evidence. To explicitly evaluate dynamic spatial memory, we manually annotate repeated queries at different timestamps with updated answers (see Figure 1).

**Stage 3: Multiple-choice Conversion.** We generate multiple-choice options via a multi-stage distractor pipeline that combines retrieval-based candidates, neural filtering, and LLM refinement. To remove low-quality distractors and ensure a unique correct answer, we apply an off-the-shelf Natural Language Inference (NLI) model, DeBERTa-v3-large-mnli (He et al., 2020), which predicts entailment relations between sentence pairs, as a logical-consistency filter. After each round, we check pairwise entailment among all options, discard any logically overlapping distractors, and regenerate replacements with the pipeline. Details of this procedure are provided in § F.4.

*Table 2.* Comparison with egocentric VQA and video spatial intelligence benchmarks. Dynamic: whether spatial relations change over time. U-Centric: User-centric spatial relations. Manual: QA pairs are manually annotated. Stream: online streaming QA setting. In&Out: Videos include both indoor and outdoor scenes. Evidence: Temporal or descriptive evidence for each QA.

| Benchmark | # QAs. | # Vid | VLen. | Spatial | Dynamic | U-Centric | Manual | Stream | In&Out | Evidence |
|---|---|---|---|---|---|---|---|---|---|---|
| *Egocentric VQA* | | | | | | | | | | |
| QAEgo4D (Grauman et al., 2022) | 1.8K | 166 | 8m | ✗ | ✗ | ✗ | ✓ | ✗ | ✓ | ✓ |
| EgoSchema (Mangalam et al., 2023) | 5K | 5K | 3m | ✗ | ✗ | ✗ | ✗ | ✗ | ✓ | ✗ |
| HourVideo (Chandrasegaran et al., 2024) | 13K | 500 | 46m | ✗ | ✗ | ✗ | ✗ | ✗ | ✓ | ✗ |
| EgoLifeQA (Yang et al., 2025a) | 6K | 6 | 44h | ✗ | ✗ | ✗ | ✗ | ✓ | ✓ | ✗ |
| EgoBlind (Xiao et al., 2025) | 5.3K | 1.3K | 40s | ✗ | ✗ | ✗ | ✗ | ✓ | ✓ | ✗ |
| *Video Spatial Intelligence* | | | | | | | | | | |
| VSI-Bench (Yang et al., 2025b) | 5K | 300 | 2m | ✓ | ✗ | ✗ | ✗ | ✗ | ✗ | ✗ |
| STI-Bench (Li et al., 2025c) | 2K | 300 | 2m | ✓ | ✓ | ✗ | ✗ | ✗ | ✓ | ✗ |
| OST-Bench (Lin et al., 2025c) | 10K | 1.4K | 35s | ✓ | ✓ | ✗ | ✗ | ✓ | ✗ | ✗ |
| MMSI-Video (Lin et al., 2025b) | 1.1K | 1.3K | 72s | ✓ | ✓ | ✗ | ✓ | ✗ | ✓ | ✗ |
| VSI-Super (Yang et al., 2025e) | 700 | 300 | 92m | ✓ | ✗ | ✗ | ✗ | ✓ | ✗ | ✗ |
| UCS-Bench (Ours) | 8.1K | 532 | 20m | ✓ | ✓ | ✓ | ✓ | ✓ | ✓ | ✓ |

**Stage 4: Quality control.** We use a two-stage quality-control pipeline combining model-based bias checks and human auditing. First, we run blind-choice tests (answering without video) with GPT-5 (Singh et al., 2025) to detect and revise options where the correct option can be identified by language priors. Human reviewers then verify whether each QA pair (i) concerns user-centric location or orientation, (ii) has aligned timestamps and evidence spans, and (iii) uses a clear egocentric perspective with unambiguous referents. More details are given in § F.5.

## 3.2. Dataset Statistics.

UCS-Bench contains 8,114 QA pairs grounded in 532 diverse egocentric videos (typical visual scenes are shown at Appendix § 16). The QA pairs are organized into four high-level categories, each with two fine-grained sub-categories (Figure 2), exhibiting a balanced yet functionally distinct distribution. Object Recall & Quantity Tracking constitutes the largest portion (34.39%), followed by Ego-centric Position & Orientation Memory (25.73%), Embodied Trajectory & Movement Memory (21.73%), and Perceived Proximity & Reachability Memory (18.15%). This distribution reflects the relative difficulty and prevalence of different spatial reasoning demands in long-horizon egocentric settings. Other QA statistics are provided in § F.2.

We further analyze the temporal distributions of the questions and answer evidence from three dimensions: *Evidence Question Interval* measures the time gap between a question and the endpoint of its supporting evidence, i.e., the last time the queried object appears. *Evidence Span* measures how long an answer's evidence spans. *Repeated Question Interval* analyzes the time gap between two repeated questions. Figure 4 shows that more than 40% of questions require evidence observed more than 30 seconds before the query moment, indicating the need for long-term memory. Moreover, the evidence often spans long temporal intervals, indicating long-duration tracking and accumulating user movements for answering. The repeated questions appear from seconds

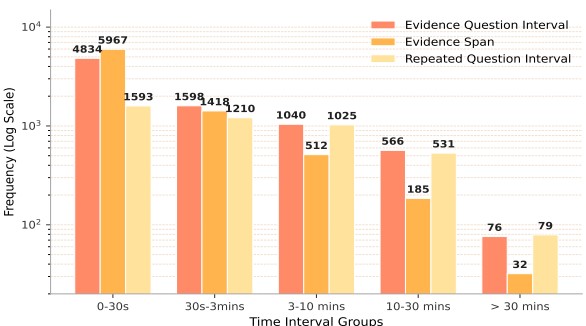

*Figure 4.* Distribution of QA temporal characteristics, all underscoring the importance of long-term and evolving spatial memory.

to over 30 minutes away from their corresponding original questions, enabling consistency evaluation across diverse temporal horizons.

## 3.3. Dataset Comparison.

Table 2 compares UCS-Bench with prior egocentric VQA and video spatial intelligence benchmarks along essential axes. Egocentric VQA targets semantic understanding and typically lacks explicit spatial focus. While video spatial intelligence emphasizes spatial reasoning, it is often presented from a third-person perspective and focuses on offline video understanding, missing online user-centric spatial relations by engaging with camera wearers. Moreover, they often lack manual annotation, broad indoor & outdoor coverage, and answer evidence labels. In contrast, UCS-Bench is the only benchmark in Table 2 that satisfies *all* listed properties, enabling long-horizon evaluation of reasoning about spatial state changes relative to the user's real-time movement. We also provide more visual comparisons in Figure 8.

## 4. Method

### 4.1. Solution Overview.

To correctly answer the questions in UCS-Bench, it is important to dynamically track and link the objects to the user's real-time location/orientation. This highlights the

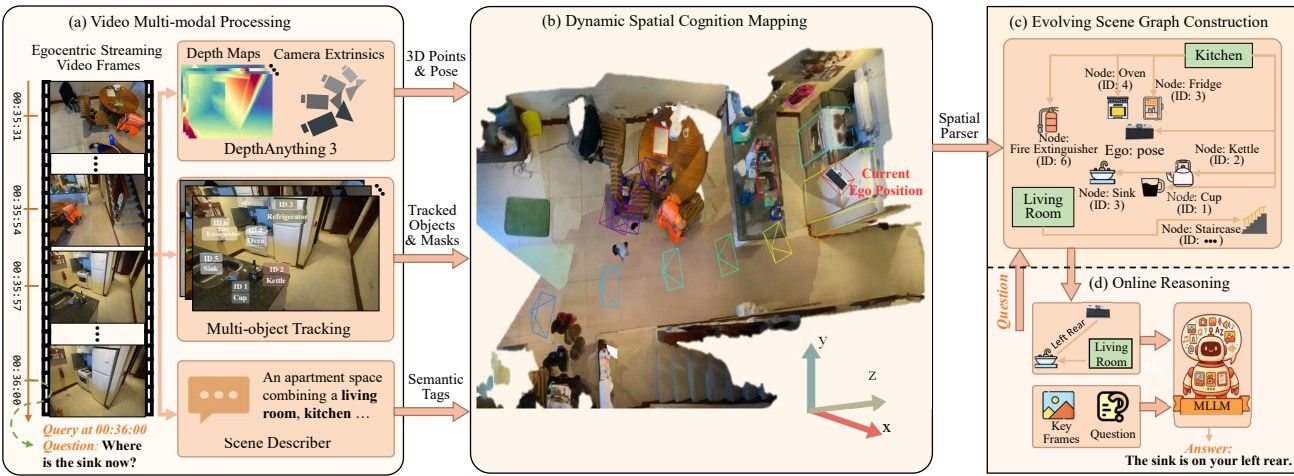

*Figure 5.* Overview of DirectMe. For the incoming video stream, (a) we extract multimodal cues which are then (b) fused to build a global metric-semantic map with the ego position. (c) We follow the map to construct an evolving scene graph that maintains the evolving ego pose and spatial relations among tracked objects and the user. (d) For online QA, we retrieve question-relevant key frames and a minimal subgraph as the prompt for MLLM to answer the question.

need to construct and maintain an evolving scene graph that memorizes the objects' relative location/orientation to the user at each timestamp. We therefore propose **DirectMe**, a streaming QA framework that maintains a real-time, camera pose-anchored spatial scene graph as a continually updated memory module for user-centric spatial intelligence. Unlike offline video QA, where the entire video is available for answering, DirectMe operates under a causal streaming setting: at any question time $t$, the answer can only depend on observations and scene-graph states accumulated up to $t$. The scene-graph construction itself is question-independent and can thus be performed in the background whenever no question is triggered. Rather than building a static graph over the full video, DirectMe incrementally updates a fixed-window scene graph along the temporal stream, anchoring incoming observations to camera poses in a consistent reference frame. To support robust continual accumulation, the memory module leverages the foundation geometry model (Lin et al., 2025a) to infer depth and camera extrinsics, making the resulting memory resilient to viewpoint rotations, blur, occlusions, and indoor–outdoor appearance shifts.

As shown in Figure 5, the pipeline consists of background video-to-structured-memory processing (a–c) and online question answering (d). During background processing, DirectMe continuously extracts geometric, pose, and semantic signals from the incoming stream, and integrates them over time to incrementally update the scene-graph memory. This memory maintains a structured representation of the evolving environment, including relevant objects, agents, and their spatial-temporal relations. When a question is asked at time $t$, the online QA module retrieves only from the memory available up to $t$, ensuring causal access to past observations. It then selects a minimal question-conditioned

subgraph and a set of keyframes, which are fed into the MLLM for response generation.

## 4.2. Offline Spatial Scene-Graph Memory

*(1) Video multi-modal processing.* We process the input video at 1 fps, denoted as $\{\mathbf{I}(t)\}_{t=1}^{T}$. For each frame, we estimate metric depth and camera pose using DepthAnything3 (Lin et al., 2025a), obtaining depth maps $\{\mathbf{D}(t)\}_{t=1}^{T}$ and an ego-pose trajectory $\{\mathbf{T}_{w \leftarrow e}(t)\}_{t=1}^{T}$. To extract object-level observations, we first run Grounding-DINO (Liu et al., 2024b) with an Objects365 category vocabulary to detect candidate objects in each frame. The resulting detections are then used to initialize SAM2 (Ravi et al., 2025), which propagates object masks over time and produces temporally consistent instance tracks. This yields a set of persistent object instances $\mathcal{I} = \{1, \dots, N\}$, with per-frame masks and bounding boxes $\{\mathbf{M}_i(t) \in \{0,1\}^{H \times W}\}_{i \in \mathcal{I}, t=1}^{T}$ and $\{\mathbf{b}_i(t) \in \mathbb{R}^4\}_{i \in \mathcal{I}, t=1}^{T}$. Finally, a lightweight MLLM-based scene describer (Bai et al., 2025a) assigns frame-level semantic tags $\{y(t) \in \mathcal{Y}\}_{t=1}^{T}$, where $\mathcal{Y}$ denotes the predefined scene taxonomy. Further implementation details are provided in Appendix § D.1.

*(2) World-aligned metric-semantic mapping.* We integrate these multi-modal signals into a global *metric-semantic* map $\mathcal{M}$ defined in a shared world coordinate frame. The map is initialized from the first observation ($t{=}1$) and incrementally updated over time, enabling temporally consistent localization, spatial reasoning, and semantic grounding across long-horizon video sequences. Such a world-aligned representation provides unified support for downstream tasks that require coherent tracking of object-level semantics and geometry under dynamic scene evolution. Further implementation details are provided in Appendix § D.2.

*(3) Evolving scene-graph construction.* Given $\mathcal{M}$ (with world-frame object anchors) and the ego-pose trajectory $\{\mathbf{T}_{w \leftarrow e}(t)\}_{t=1}^{T}$, we perform spatial parsing in the world coordinate system to build a scene-graph memory $\mathcal{G} = (\mathcal{V}, \mathcal{E})$. The node set $\mathcal{V}$ includes persistent object nodes grounded by their global 3D locations and semantic place nodes representing map-induced regions, while edges $\mathcal{E}$ encode object–object and object–place relations computed from world-frame geometry. At query time $t$, relations are rendered in the current egocentric frame via $\mathbf{T}_{e \leftarrow w}(t) = \mathbf{T}_{w \leftarrow e}(t)^{-1}$ for pose-conditioned reasoning. Construction details are provided in Appendix § D.3.

### 4.3. Online Video Reasoning

At query time $t$, we first parse the question into a structured query intent, specifying target objects, semantic attributes, and requested reasoning primitives (e.g., counting, localization). Based on this intent, we retrieve a compact query-conditioned subgraph from the global scene graph, restricting candidate nodes to those observed before $t$ to respect temporal causality. Candidate nodes are scored and ranked by matching the query intent against their semantic labels, visual attributes, and scene tags, retaining the top-$K$ instances alongside their supporting keyframes.

Next, we transform the world-frame coordinates of each retrieved object into the current egocentric frame via $T_{e \leftarrow w}(t) = T_{w \leftarrow e}(t)^{-1}$, yielding camera-to-object relation edges that encode relative direction, metric distance, and reachability. Providing this egocentric, pose-aware context to the MLLM ensures that spatial predictions remain geometrically verifiable, enabling accurate reasoning even for out-of-view targets (e.g., "on your left rear" in Figure 5 (d)).

## 5. Experiments

### 5.1. Experimental Settings

**Evaluated Models.** We evaluate 18 representative MLLMs (4B-38B) spanning major paradigms: **(i)** *proprietary* models (DeepSeek-V3 (Liu et al., 2024a), Doubao-seed-1.6V, GPT-5 (Singh et al., 2025), Gemini3-Flash (Google, 2025)); **(ii)** strong *general-purpose* open models (Qwen2.5/3-VL (Bai et al., 2025b;a), InternVL3/3.5 (Zhu et al., 2025a; Wang et al., 2025a), LLaVA-Video (Zhang et al., 2025c), Intern-Video2.5 (Wang et al., 2025b)); **(iii)** *spatial-centric* models (Spatial-MLLM (Wu et al., 2025a), ViLaSR (Wu et al., 2025b), SenseNova-SI (Cai et al., 2025b)); and (iv) *streaming/online* models (Flash-VStream (Zhang et al., 2025a), VideoChat-Online (Huang et al., 2025), Dispider (Qian et al., 2025)). This selection enables a controlled comparison across parameter scales and helps diagnose whether improvements stem from general multimodal capacity, explicit spatial priors, or streaming-specific design, rather than from model size alone. It also facilitates a more interpretable comparison across settings and helps analyze how different architectural choices influence egocentric spatial reasoning performance.

### 5.2. Main Results

**General Observations.** Existing SOTA results on UCS-Bench barely reach around 50%, whereas humans achieve an accuracy of 91%, revealing a substantial performance gap of ∼40%. Surprisingly, advanced general-purpose MLLMs often outperform both spatially aware models and models specifically designed for long-form streaming video QA. This indicates that strong general visual–language reasoning and instruction-following capabilities can be more critical than specialized architectural designs for generalized spatial modeling (*e.g.*, user-centric and dynamic) or long-context processing on UCS-Bench. Our method achieves the best overall accuracy and leads most spatially demanding categories, although some semantic-recognition subcategories remain stronger for large general-purpose baselines. Nevertheless, the large gap to human performance suggests that current models still struggle with sustained ego-grounded understanding, long-range tracking, and precise temporal localization and evidence accumulation, highlighting UCS-Bench as a challenging problem far from solved.

**Object Motion *vs.* Ego-Motion.** Models consistently perform better on *object-trajectory* than on *ego-trajectory* questions, exposing a systematic weakness in ego-motion reasoning that is largely overlooked by existing spatial benchmarks. Rather than maintaining explicit ego-motion estimates, current MLLMs appear to rely on readily available pixel-level cues such as object displacement. This limitation is amplified in questions that couple object dynamics with ego-motion, especially those involving relative position and orientation, where accuracy drops markedly compared to single-factor egocentric tracking (*Trajectory & Movement*). Overall, these results indicate that existing models struggle to integrate ego-motion with long-horizon object memory into a coherent, viewpoint-conditioned spatial representation for user-centric spatial reasoning.

**Dynamic tracking remains highly challenging.** Performance drops when tasks require maintaining a cumulative state compared to recognizing momentary visual features. For example, GPT-5 performs relatively well on *Obj. Category* (58.8%), which relies on semantic recognition, but suffers a significant 25.3% drop in *Obj. Quantity* (33.5%). Furthermore, despite being a spatial task, *Position & Orientation* (37.8%) scores notably lower than *Trajectory & Movement* (44.5%). This suggests that even when models can exploit motion/trajectory cues, they struggle to maintain temporal coherence for downstream state updates, e.g., inferring final relative positions and tracking quantity changes.

*Table 3.* **Performance of different models on UCS-Bench.** The best , second-best , and third-best model results are highlighted. Note that *Proximity & Reachability* is a binary-choice task, while the others are five-way multi-choice. For fair comparison and consistency, all proprietary models are evaluated with up to 50 input images (the maximum rate of GPT-5 API).

| Models | Size | # Frame | Trajectory & Movement | | | Position & Orientation | | | Proximity & Reachability | | | Category & Quantity | | | Overall |
|---|---|---|---|---|---|---|---|---|---|---|---|---|---|---|---|
| | | | Overall | My Traj. | Obj. Traj. | Overall | Ego-centric | Relative | Overall | Distance | Reachability | Overall | Obj. Category | Obj. Quantity | |
| Human | - | - | 91.7 | 90.9 | 94.1 | 87.6 | 84.0 | 89.8 | 96.8 | 95.0 | 98.2 | 90.0 | 94.3 | 88.1 | 91.0 |
| Random | - | - | 19.1 | 20.8 | 14.2 | 22.7 | 24.4 | 21.5 | 49.7 | 51.8 | 48.1 | 19.4 | 20.3 | 19.0 | 25.7 |
| **Proprietary MLLMs** | | | | | | | | | | | | | | | |
| DeepseekV3 | - | 50 | 42.0 | 41.0 | 44.8 | 33.2 | 35.4 | 31.6 | 37.7 | 45.9 | 31.2 | 32.3 | 36.8 | 30.3 | 35.6 |
| Doubao-seed-1.6V | - | 50 | 49.6 | 49.4 | 50.1 | 42.8 | 46.4 | 40.4 | 56.0 | 54.9 | 56.8 | 44.6 | 57.0 | 39.0 | 47.3 |
| GPT-5 | - | 50 | 44.5 | 43.3 | 48.1 | 37.8 | 40.2 | 36.3 | 55.4 | 52.9 | 57.5 | 41.4 | 58.8 | 33.5 | 43.7 |
| Gemini3-Flash | - | 50 | 43.8 | 43.2 | 45.5 | 38.0 | 38.7 | 37.6 | 52.9 | 50.2 | 54.9 | 38.9 | 56.2 | 31.1 | 42.2 |
| **Open-Source MLLMs** | | | | | | | | | | | | | | | |
| InternVideo2.5 | 8B | 64 | 44.2 | 43.9 | 45.2 | 39.4 | 42.4 | 37.5 | 52.7 | 49.1 | 55.5 | 42.2 | 53.0 | 37.4 | 43.8 |
| LLaVA-Video | 7B | 64 | 40.4 | 40.9 | 38.7 | 38.7 | 42.0 | 36.6 | 47.2 | 45.9 | 48.2 | 41.8 | 52.4 | 37.1 | 41.7 |
| Qwen2.5-VL | 7B | 64 | 41.6 | 41.0 | 43.1 | 40.8 | 42.8 | 39.5 | 51.3 | 47.4 | 54.3 | 42.5 | 50.1 | 39.2 | 43.4 |
| | 32B | 64 | 50.3 | 49.4 | 53.0 | 40.9 | 43.1 | 39.5 | 55.4 | 50.4 | 59.3 | 46.0 | 56.4 | 41.4 | 47.3 |
| Qwen3-VL | 8B | 64 | 44.6 | 43.1 | 48.7 | 40.1 | 43.6 | 37.9 | 50.9 | 49.2 | 52.1 | 42.9 | 52.8 | 38.4 | 44.0 |
| | 32B | 64 | 50.3 | 49.0 | 54.1 | 46.5 | 50.1 | 44.3 | 55.2 | 53.9 | 56.3 | 46.7 | 58.8 | 41.2 | 49.0 |
| InternVL3 | 8B | 64 | 44.9 | 44.4 | 46.5 | 39.1 | 42.2 | 37.2 | 52.2 | 48.6 | 55.1 | 42.2 | 53.5 | 37.1 | 43.8 |
| | 38B | 64 | 53.7 | 52.4 | 57.3 | 45.6 | 46.9 | 44.8 | 55.3 | 54.0 | 56.3 | 45.8 | 60.4 | 39.3 | 49.2 |
| InternVL3.5 | 8B | 64 | 44.0 | 42.3 | 48.8 | 39.8 | 40.6 | 39.2 | 52.9 | 49.3 | 55.7 | 39.2 | 47.9 | 35.2 | 42.9 |
| | 38B | 64 | 50.2 | 49.5 | 52.2 | 46.3 | 45.9 | 46.6 | 53.9 | 53.5 | 54.2 | 45.0 | 56.5 | 40.1 | 48.1 |
| **Spatial Models** | | | | | | | | | | | | | | | |
| Spatial-MLLM | 3B | 64 | 33.2 | 34.1 | 30.6 | 26.1 | 28.6 | 24.4 | 53.6 | 46.0 | 59.5 | 30.9 | 36.3 | 28.5 | 34.3 |
| SenseNova-SI | 8B | 64 | 38.2 | 38.7 | 36.3 | 30.8 | 32.0 | 30.0 | 49.0 | 47.2 | 50.3 | 37.0 | 45.5 | 33.3 | 37.8 |
| VG-LLM | 8B | 64 | 34.6 | 35.7 | 31.3 | 35.0 | 38.3 | 32.4 | 51.9 | 45.9 | 56.6 | 35.9 | 47.3 | 30.9 | 38.3 |
| ViLaSR | 7B | 64 | 37.3 | 38.1 | 34.9 | 33.2 | 34.5 | 32.3 | 52.3 | 46.0 | 57.2 | 40.7 | 48.7 | 37.1 | 40.1 |
| Cambrian-S | 7B | 64 | 44.2 | 44.0 | 44.5 | 35.9 | 36.2 | 35.7 | 52.3 | 43.7 | 59.0 | 40.8 | 50.4 | 36.6 | 42.4 |
| **Streaming Models** | | | | | | | | | | | | | | | |
| Flash-VStream | 7B | 1fps | 26.1 | 27.3 | 22.1 | 23.0 | 23.7 | 22.5 | 48.8 | 44.4 | 52.2 | 23.4 | 27.2 | 21.8 | 28.5 |
| Dispider | 7B | 1fps | 40.2 | 40.6 | 38.7 | 29.1 | 30.1 | 28.4 | 51.6 | 49.2 | 53.5 | 40.2 | 38.7 | 29.4 | 36.7 |
| VideoChat-Online | 4B | 1fps | 35.7 | 36.3 | 33.7 | 30.3 | 32.2 | 29.0 | 51.9 | 45.7 | 56.8 | 36.4 | 43.1 | 33.5 | 37.5 |
| DirectMe (w/ InternVL3) | 8B | 1fps | 52.6 | 51.6 | 55.4 | 47.1 | 47.3 | 47.0 | 56.9 | 54.2 | 59.1 | 46.0 | 54.5 | 42.2 | 49.7 |
| DirectMe (w/ Qwen3-VL) | 8B | 1fps | 54.1 | 53.0 | 57.2 | 47.6 | 50.2 | 46.0 | 58.0 | 55.7 | 59.8 | 46.4 | 53.7 | 43.1 | 50.5 |

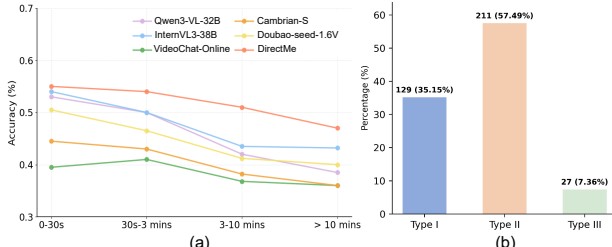

*Figure 6.* (a) Accuracy trends across different time intervals between Evidence Question Interval. (b) Distribution of error categories. SRF: Spatial Reasoning Failure. TSTF: Temporal State Tracking Failure. VMA: Visual Memory Amnesia.

This reveals a fundamental weakness in updating latent spatial states over time.

**Robustness to Question–Evidence Distance.** Figure 6 (a) shows that our method consistently achieves higher accuracy, while all models degrade as the temporal gap between the question and its answer evidence increases. Notably, our performance declines more gradually than other baselines, especially over long intervals (>3 minutes). This indicates that our scene graph memory helps reduce model sensitivity to temporal distance, as it better preserves historical spatial evidence, which benefits long-horizon reasoning.

## 5.3. Error Analysis

To gain a deeper understanding, we conduct two complementary studies. The first is a benchmark-level analysis that identifies common failure patterns shared by existing VLMs, as illustrated in Figure 6 (b). The second is a method-level root-cause analysis that diagnoses where failures arise

*Table 4.* Root-cause analysis of 500 failed QA cases.

| Primary failure source | Ratio (%) |
|---|---|
| Tracking failure | 40 |
| Ego-pose relation failure | 23 |
| Retrieval failure | 20 |
| Reasoning failure on correct evidence | 17 |

within our proposed DirectMe pipeline, as shown in Table 4. The uncovered failure cases span all four UCS-Bench task categories, revealing systematic limitations shared by current models. Specifically, we group these failures into three dominant error types, which reflect progressively more complex spatial reasoning requirements. We also present visual examples of these three types in Figure 15 of the Appendix.

*(i) Spatial Reasoning Failure.* Errors arising from incorrect 3D geometric understanding or egocentric spatial relation judgments, such as misinterpreting relative directions or spatial layouts from the user's perspective.

*(ii) Temporal State Tracking Failure.* Errors arising from the inability to consistently maintain object states over time—such as failures in trajectory reconstruction, temporal ordering, and event counting-indicate limited temporal coherence. These errors account for over 55% of all observed failures, highlighting a substantial gap between nominal long-context capacity and effective temporal reasoning.

*(iii) Visual Memory Amnesia.* Errors caused by forgetting visual details observed earlier in the video stream, particularly when targets exit and later re-enter the field of view.

To identify the bottlenecks in our streaming memory pipeline, we analyze 500 failed QA cases from DirectMe.

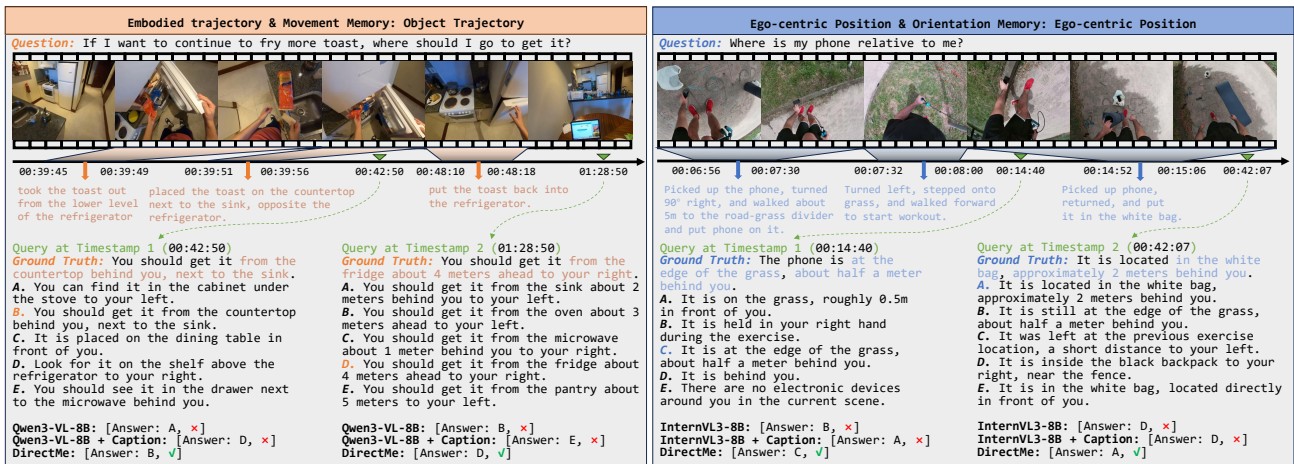

*Figure 7.* Qualitative results on UCS-Bench across **indoor and outdoor** scenarios, using Qwen3-VL-8B and InternVL3-8B as baselines.

*Table 5.* Ablation study on our method. Description denotes the MLLM-generated spatial description, while Graph denotes the retrieved results from our scene-graph memory.

| Models | Desc. | Graph | Traj. & Mov | Pos & Ori | Prox & Reach | Rec & Qua | All |
|---|---|---|---|---|---|---|---|
| Qwen3-VL (baseline) | - | - | 44.6 | 40.1 | 50.9 | 42.9 | 44.0 |
| | ✓ | - | 44.3 | 41.9 | 52.1 | 42.5 | 44.5 |
| | - | ✓ | 54.6 | 46.3 | 56.8 | 46.7 | 50.2 |
| **DirectMe** | ✓ | ✓ | 54.1 | 47.6 | 58.0 | 46.4 | 50.5 |

Each case—comprising the QA pair, time-stamped video, and retrieved subgraph text - was independently categorized into a single primary root cause by human annotators to eliminate cascading bias. As summarized in Table 4, tracking (40%) and ego-pose relation errors (23%) constitute the main failure modes, outnumbering pure retrieval (20%) and reasoning deficits (17%). This distribution indicates that the primary challenge has shifted from information retrieval to maintaining a temporally consistent, user-centric spatial state. Consequently, robust object persistence and viewpoint-aligned memory updates emerge as the core bottlenecks for streaming egocentric QA.

### 5.4. Model Ablation

In Table 5, we isolate the effect of *Desc.* versus *Graph* memory. *Desc.* augments the Qwen3-VL-8B baseline by adding per-frame scene-level spatial descriptions generated by themselves (see § E.2) as extra context alongside the original frames and question. This yields only marginal gains, suggesting that third-person view descriptions fail to capture the *explicit*, viewpoint-conditioned relations needed for out-of-sight, past-to-now queries. In contrast, *Graph* maintains an explicit pose-anchored scene graph of object locations, enabling direct retrieval of spatial relations at query time. Figure 7 suggests that this structured memory resolves challenging cases where the caption-based variant fails, leading to consistent improvements.

## 6. Conclusion

In this work, we study user-centric continual spatial intelligence in egocentric video streams and introduce **UCS-Bench**, a carefully constructed egocentric QA benchmark. We further propose **DirectMe**, a streaming QA framework that highlights a dynamic scene graph as spatial memory to track objects and their relations under users' continuous movement. Extensive experiments show that current MLLMs remain far from solving this task, while DirectMe consistently outperforms strong baselines, with advantages that grow as the temporal gap between question and evidence increases. The findings demonstrate our method's effectiveness and signal UCS-Bench as a significant challenge and a new avenue for advancing future MLLMs.

## Impact Statement

This work introduces a dataset and benchmark, as well as a corresponding method for advancing research on user-centric continual spatial intelligence in egocentric video streams. The primary goal of this work is to support the development and evaluation of machine learning models capable of online long-term dynamic spatial reasoning under ego-motion, which may benefit applications such as assistive technologies and embodied AI systems.

As a dataset and methodology contribution, this work does not involve direct deployment in real-world decision-making systems. Potential ethical considerations mainly relate to the responsible use of egocentric visual data, including privacy and data governance. We mitigate such risks by adhering to established data collection and anonymization practices and by restricting the dataset to research purposes. We do not foresee significant negative societal impacts arising directly from this work beyond those commonly associated with advances in machine learning research.

## Acknowledgments

This research is supported by the Ministry of Education, Singapore, under its MOE Academic Research Fund Tier 2 (MOE-T2EP20125-0037).

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

## Supplementary Material for UCS-Bench

## A. Overview

This supplementary material provides additional details and analyses to complement the main paper. We include:

1. Spatial Intelligence Benchmark Comparison (§B) presents a high-level comparison of spatial intelligence evaluation along three dimensions: *Static Spatial Relation*, *Dynamic Spatial Relation*, and *User-Centric Spatial Relation* (Figure. 8). Representative benchmarks for each dimension are respectively VSI-Bench (Yang et al., 2025b), MMSI-Video-Bench (Yang et al., 2025d), and the proposed UCS-Bench, enabling a structured comparison across progressively complex spatial reasoning settings.

2. Implementation Details (§C) includes the implementation specifics for evaluation based on pre-extracted video frames at 1 fps, highlighting the comparison between Uniform Sampling (which captures global history) and Streaming Window Sampling (which focuses on recent context). Additionally, the number of sampled frames K and image resolution are treated as hyperparameters to explore performance under various configurations.

3. Processing methods (§D.1) extracts multi-modal signals from egocentric videos, including per-frame depth maps and ego poses using DepthAnything3 (Lin et al., 2025a), persistent object masks and trajectories via Grounding-DINO (Liu et al., 2024b) or YOLO-World (Cheng et al., 2024b) and SAM2 (Ravi et al., 2025), and coarse scene semantics using Qwen3-VL (Bai et al., 2025a), serving as inputs for constructing the global metric-semantic map.

4. Evaluation methods (§D.2-§D.3) describes a spatial reasoning evaluation method that integrates visual frames, camera poses, object tracking, and scene tags into a global map, supporting both offline and real-time reasoning.

5. Implementation Parameters (§D.4) describes the experimental hardware setup, the model implementation framework, and the inference parameter settings.

6. Prompt (§D.5) defines a set of prompt templates for egocentric video question answering (Video QA) tasks, used to guide the model in generating answers under different strategies (Uniform Sampling, Stream Sampling, Spatial Captions, Scene Graph, etc.).

7. Hyperparameter Comparisons (§E.1) provides a detailed analysis of the impact of key hyperparameters on model performance, including model size, spatial resolution, and temporal context length.

8. Spatial Captions (§E.2) augments visual inputs with frame-level spatial captions generated by a VLM, which explicitly describe object positions, orientations, and relative layouts. These captions are concatenated with the corresponding frames and jointly fed into the model to provide additional explicit spatial information during inference.

9. Dataset Source (§F.1) collects data from several publicly available datasets, including EgoLife (Yang et al., 2025a), HourVideo (Chandrasegaran et al., 2024), ScanNet (Dai et al., 2017), EPIC-KITCHENS (Damen et al., 2020), EgoBlind (Xiao et al., 2025), and TeleEgo (Yan et al., 2025). The collected data focuses on user-centric continual video, and covers a diverse set of categories, including egocentric daily activities, long-duration videos, indoor 3D scenes, fine-grained object interactions, assistive scenarios, and continuous streaming perception.

10. Dataset Statistics (§F.2) includes four categories of question templates—ego-centric spatial memory, trajectory movement memory, proximity reachability judgment, and object recall quantity tracking—designed to evaluate models on dynamic spatial relationships and temporal reasoning. Word cloud analysis of questions and answers highlights frequent terms such as direction, relative position, distance, and temporal markers, indicating that models must handle long-term memory, precise spatial localization, and continuous reasoning over evolving scenes.

11. Annotation System (§F.3) comprises the video QA annotation tool and the QA refinement verification workflow. It enables annotators to create temporally and spatially grounded question-answer pairs, apply perspective correction, optimize answer options (including AI-assisted distractor generation and counting question heuristics), and conduct blind-test adversarial validation to ensure high-quality, semantically accurate, and bias-resistant data through a combination of human verification and AI assistance.

12. Generation Distractor Pipeline (§F.4) includes question type categorization (binary, count, multi-choice) and a multi-stage distractor generation process. The pipeline combines retrieval-based candidate extraction, multi-criteria validation, diversity-aware selection, LLM-based refinement, and NLI-guided iterative optimization to produce high-quality, semantically plausible, and diverse distractors.

13. Quality-control Pipeline (§F.5) provides an overview of our procedure to ensure high-quality distractor options and reliable question-answer pairs. It covers automated evaluation using GPT-5 to identify and improve ineffective distractors, addresses semantic and information balance issues, and incorporates manual inspection to verify temporal evidence alignment, answer clarity, and relevance to dynamic spatial relationships.

14. Human Performance and QA (§F.6) evaluates human upper-bound performance on the benchmark. The study aligns tasks with model questions, allows full-context video access, enforces independent answering by experts and reports average accuracy as a reference for model evaluation.

15. Representative Failure Cases (§G.1) includes illustrative examples for Type I (Spatial Reasoning), Type II (Temporal State Tracking), and Type III (Visual Memory Amnesia) errors, highlighting typical failure patterns and underlying causes.

16. Insights and Implications (§G.2) includes analysis of core capability gaps in current VLMs, covering lack of 3D spatial grounding, state memory bottleneck, and long-range visual forgetting, as well as potential directions for improvement.

17. Qualitative Analysis (§G.3) includes representative visual scenes from diverse environments, demonstrating dataset breadth and ecological validity for evaluating spatial reasoning and memory in daily-life contexts.

Together, these materials support reproducibility and provide deeper insight into the benchmark design and the proposed pose-anchored memory framework.

## B. Spatial Intelligence Benchmark Comparison

The evaluation of spatial intelligence in multimodal models can be categorized into three progressive dimensions: Static Spatial Relation, Dynamic Spatial Relation, and User-Centric Spatial Relation, as illustrated in Figure 8. While earlier evaluations often conflated these capabilities, distinguishing between them allows for a more granular understanding of a model's internal world model. This section delineates these dimensions and examines how current benchmarks, specifically VSI-Bench (Yang et al., 2025b), MMSI-Video-Bench (Yang et al., 2025d), and our proposed UCS-Bench, serve as evaluative frameworks for each.

**Static Spatial Relation** focuses on the fundamental "thinking" mechanisms of spatial awareness within fixed environments. This dimension prioritizes internal spatial simulation, requiring models to perform multi-step mental rotation, deduce hidden geometric properties, and resolve abstract relationships between stationary objects. VSI-Bench (Yang et al., 2025b) serves as a foundational representative for this category. By emphasizing cognitive bottlenecks in structured scenes, it effectively measures a model's capacity for logical spatial deduction. However, static relations often operate within a "third-person" or "God's eye" view, which may not fully capture the fluidity of real-world interactions.

**Dynamic Spatial Relation** extends these capabilities into the temporal domain, shifting the focus toward spatial consistency and perception within moving sequences. This dimension evaluates how models track objects, maintain orientation across continuous frames, and predict trajectories, demanding a sophisticated fusion of temporal reasoning and spatial localization. MMSI-Video-Bench (Yang et al., 2025d) is the primary benchmark addressing this need. While it excels at capturing motion-driven spatial intelligence, its reliance on continuous video streams often overlooks the deep structural mapping of high-resolution complex scenes or the ability to perform non-linear, cross-view synthesis.

**User-Centric Spatial Relation** represents the most complex dimension, requiring a holistic integration of high-fidelity scene understanding with ego-centric perspective synthesis. Unlike static logic or linear motion tracking, user-centric spatiality demands that the model builds a comprehensive global world model from disparate, non-continuous visual cues, often centered around the observer's relative position and intent. To address this, we introduce **UCS-Bench**, which acts as a unifying stress test. By bridging the gap between abstract logic and embodied perception, UCS-Bench measures general-purpose spatial intelligence through tasks that require simultaneous excellence in fine-grained localization, structural mapping, and complex environment navigation from a user-oriented perspective.

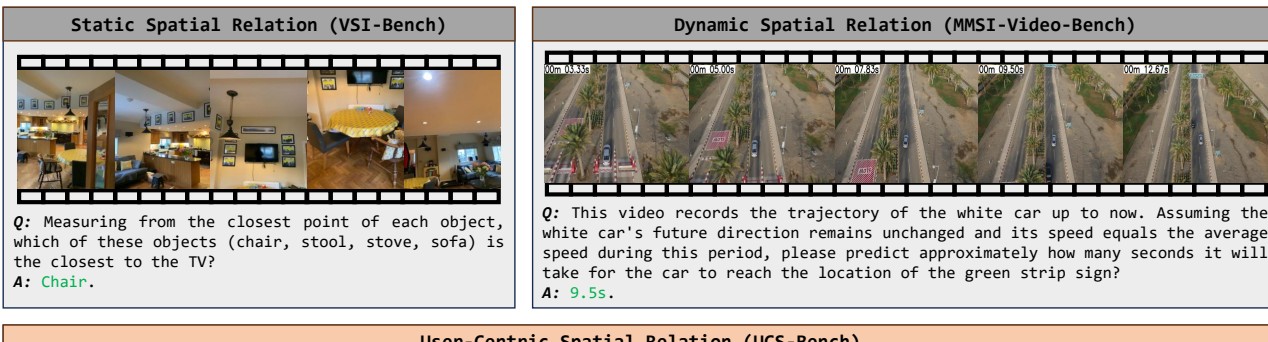

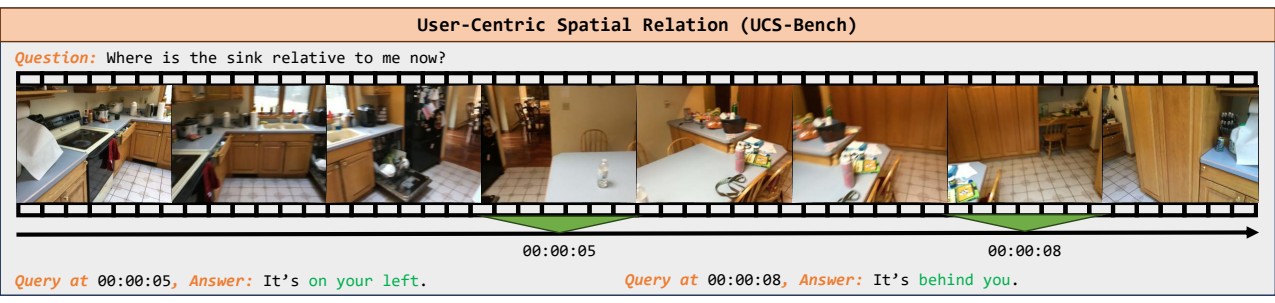

*Figure 8.* Comparison of Spatial Reasoning Benchmarks. This figure compares different spatial reasoning benchmarks: VSI-Bench, which focuses on static spatial relations, such as evaluating distances or relative positions between objects (e.g., a chair and a TV); MMSI-Video-Bench, which emphasizes dynamic reasoning, including trajectory tracking and time-to-arrival prediction for moving objects; and UCS-Bench, which targets egocentric, user-centric spatial understanding, where object locations (e.g., a sink) are defined relative to a moving observer's perspective.

## C. Implementation Details

In this section, we provide detailed implementation descriptions for the additional evaluation methods presented in the main paper. We compare two distinct sampling approaches, uniform sampling and sampling the most recent frames.

**Implementation.** This method utilizes pre-extracted video frames sampled at 1 frame per second, as same with (Wang et al., 2025c). For a given question occurring at timestamp $(t_q)$, we utilize the historical visual context available up to $(t_q)$. We implement two temporal sampling strategies to select $(K)$ frames as input to the Vision-Language Model:

1. **Uniform Sampling:** We uniformly sample $K$ frames from the entire history range $[0, t_q]$. This strategy captures the global context of the video but may lose high-frequency details needed for short-term dynamics. We employ prompt template 1 for this setting.

2. **Streaming Window Sampling:** We select the most recent $K$ frames within a defined temporal window $[t_q - W, t_q]$, where $W$ is the window size in seconds. If the number of available frames in the window is less than $K$, all available frames are used. This strategy focuses on the immediate temporal context relevant to the query. We employ prompt template 2 for this setting.

**Hyperparameters.** We treat the number of sampled frames $K$ and image resolution as hyperparameters, exploring various configurations in our experiments.

## D. Details of the proposed DirectMe

### D.1. Step 1: Video multi-modal processing.

We describe the implementation details of Stage 1, which extracts multi-modal signals from an egocentric video to support the downstream global metric-semantic mapping. Given an egocentric video, we uniformly sample frames at 1 FPS, forming an image stream $\{\mathbf{I}(t)\}_{t=1}^{T}$.

**Depth and ego pose.** For each sampled frame, we estimate metric depth and camera motion using DepthAnything3 (Lin

et al., 2025a) using the DA3NESTED-GIANT-LARGE-1.1 model. We run the model in a streaming manner over $\{\mathbf{I}(t)\}$ to obtain per-frame depth maps $\{\mathbf{D}(t)\}_{t=1}^{T}$ and an ego-pose trajectory $\{\mathbf{T}_{w\leftarrow e}(t)\}_{t=1}^{T}$, where we initialize the global/world reference frame at the first frame by setting $\mathbf{T}_{w\leftarrow e}(1) = \mathbf{I}$. This convention defines a consistent world coordinate system for the entire sequence, and all subsequent poses are expressed relative to this initial reference.

**Open-vocabulary detection with named mask tracks.** We obtain persistent *named* object tracks by combining Grounding-DINO or Yolo-world detection with SAM2 mask propagation. At each frame $t$, YOLO-World (Cheng et al., 2024b) is queried with an Objects365 category vocabulary and produces detections

$$\mathcal{D}(t) = \{(\mathbf{b}_m(t), \hat{c}_m(t), s_m(t))\}_{m=1}^{M_t}, \tag{1}$$

where $\mathbf{b}_m(t)$, $\hat{c}_m(t) \in \mathcal{C}_{365}$, and $s_m(t)$ denote the bounding box, predicted object name, and confidence score, respectively. These detections are used to initialize SAM2 (Ravi et al., 2025), which propagates instance masks over time and produces persistent tracks $\mathcal{I} = \{1, \ldots, N\}$ with per-frame masks $\mathbf{M}_i(t)$, boxes $\mathbf{b}_i(t)$, and visible frame sets $\mathcal{T}_i$. Thus, object names are inherited from YOLO-World, while temporal consistency is provided by SAM2.

For each track, we obtain a stable semantic name by confidence-weighted voting over the YOLO-World labels associated with its frames:

$$\hat{c}_i = \arg\max_{c \in \mathcal{C}_{365}} \sum_{t \in \mathcal{T}_i} \mathbb{I}[\hat{c}_i(t) = c] \cdot w_i(t), \tag{2}$$

where $w_i(t)$ is the detection confidence, optionally combined with mask quality. The resulting named mask tracks are used for world-frame anchoring and metric-semantic scene-graph construction.

**Scene semantics.** In parallel, we attach coarse semantic context by running a lightweight MLLM-based scene describer, Qwen3-VL-1B (Bai et al., 2025a), on each frame to generate a scene-level tag $y(t) \in \mathcal{Y}$ (e.g., kitchen, living room, street), producing $\{y(t)\}_{t=1}^{T}$. Collectively, these extracted signals-$\{\mathbf{D}(t)\}$, $\{\mathbf{T}_{w\leftarrow e}(t)\}$, $\{\mathbf{M}_i(t), \mathbf{b}_i(t)\}$, and $\{y(t)\}$, serve as the inputs to the subsequent Stage 2 for constructing the global metric-semantic map $\mathcal{M}$.

### D.2. Step 2: Global Metric–Semantic Map Fusion

Given the 1 FPS egocentric stream $\{\mathbf{I}(t)\}_{t=1}^{T}$ and the multi-modal outputs from Step 1: metric depth maps $\{\mathbf{D}(t)\}_{t=1}^{T}$, ego poses $\{\mathbf{T}_{w\leftarrow e}(t)\}_{t=1}^{T} \in SE(3)$ (with the world frame initialized at $t=1$), tracked instances $\mathcal{I} = \{1, \ldots, N\}$ with masks $\{\mathbf{M}_i(t)\}$ and boxes $\{\mathbf{b}_i(t)\}$, and frame-level scene tags $\{y(t) \in \mathcal{Y}\}$, we fuse them into a global metric-semantic map $\mathcal{M}$ in a consistent world coordinate system. Let $\pi^{-1}(\cdot)$ denote back-projection using camera intrinsics $\mathbf{K}$. For each pixel $\mathbf{u} = (u, v)$ with depth $d = \mathbf{D}(t)[\mathbf{u}]$, we recover an ego-frame 3D point

$$\mathbf{x}_e(t, \mathbf{u}) = \pi^{-1}(\mathbf{u}, d; \mathbf{K}) \in \mathbb{R}^3, \tag{3}$$

and transform it to the world frame via the estimated pose

$$\mathbf{x}_w(t, \mathbf{u}) = \mathbf{T}_{w\leftarrow e}(t)\, \tilde{\mathbf{x}}_e(t, \mathbf{u}), \tag{4}$$

where $\tilde{\mathbf{x}} = [\mathbf{x}^\top, 1]^\top$ is the homogeneous coordinate and $\mathbf{T}_{w\leftarrow e}(1) = \mathbf{I}$ under our initialization convention. Aggregating over time yields a global point set

$$\mathcal{P}_w = \bigcup_{t=1}^{T} \{\mathbf{x}_w(t, \mathbf{u})\}, \tag{5}$$

which provides a metric world reference frame consistent across viewpoint changes within the trajectory.

**Object anchoring in the world frame.** To obtain persistent object geometry aligned with tracking identities, for each instance $i \in \mathcal{I}$ we define the mask support

$$\Omega_i(t) = \{\mathbf{u} : \mathbf{M}_i(t)[\mathbf{u}] = 1\}, \tag{6}$$

lift the masked pixels to 3D, and compute a world-frame anchor (centroid) as

$$\mathbf{o}_i(t) = \frac{1}{|\Omega_i(t)|} \sum_{\mathbf{u} \in \Omega_i(t)} \mathbf{x}_w(t, \mathbf{u}) \in \mathbb{R}^3. \tag{7}$$

This yields a persistent 3D trajectory $\{\mathbf{o}_i(t)\}_{t \in \mathcal{T}_i}$ aligned with the tracker identity $i$, where $\mathcal{T}_i$ denotes the time indices at which instance $i$ is observed.

**Semantic attachment and map representation.**   We attach semantics to the global map by associating each mapped point (or its local neighborhood) with an instance id when $\mathbf{u} \in \Omega_i(t)$ and with the frame-level scene tag $y(t)$. Concretely, we maintain a global metric-semantic map

$$\mathcal{M} = \{(\mathbf{x}_j, s_j, t_j, i_j)\}_{j=1}^J, \tag{8}$$

where $\mathbf{x}_j \in \mathbb{R}^3$ is a world-frame point, $i_j \in \mathcal{I} \cup \{\varnothing\}$ denotes its associated instance (if any), $t_j \in \{1, \ldots, T\}$ is the source time index, and $s_j$ stores semantic attributes (e.g., $s_j = y(t_j)$ plus optional instance/category tags). The ego position and orientation at time $t$ are directly given by

$$\mathbf{T}_{w \leftarrow e}(t) = [\mathbf{R}(t) \mid \mathbf{p}(t)], \tag{9}$$

enabling downstream components to reference both the user's current pose and globally grounded object locations during pose-conditioned querying.

### D.3. Stage 3: Spatial Parsing and Scene-Graph Memory Construction

We construct a scene-graph memory $\mathcal{G} = (\mathcal{V}, \mathcal{E})$ by spatially parsing the global metric–semantic map $\mathcal{M}$ built in Step 2. Recall that $\mathcal{M}$ is expressed in the world frame (initialized at $t=1$) and stores world-frame 3D geometry together with semantic attributes. In particular, Step 2 provides: (i) the ego-pose trajectory $\{\mathbf{T}_{w \leftarrow e}(t) \in SE(3)\}_{t=1}^T$ with $\mathbf{T}_{w \leftarrow e}(t) = [\mathbf{R}(t) \mid \mathbf{p}(t)]$, and (ii) a set of persistent tracked instances $\mathcal{I} = \{1, \ldots, N\}$ with per-time world-frame anchors $\{\mathbf{o}_k(t) \in \mathbb{R}^3\}_{k \in \mathcal{I}, t \in \mathcal{T}_k}$, where $\mathcal{T}_k = \{t : \mathbf{M}_k(t) \text{ is valid}\}$ denotes the time indices in which instance $k$ is observed (cf. Eq. (6) in Step 2). Our goal in this stage is to convert these explicit coordinates into a compact, query-efficient memory that supports pose-conditioned spatial queries without revisiting raw frames.

**Graph definition.**   The node set $\mathcal{V}$ contains three types of nodes: object nodes $\mathcal{V}_o$, place nodes $\mathcal{V}_p$, and a single ego-anchor node $v_e$. Object nodes represent persistent entities grounded in world-frame 3D coordinates; place nodes represent semantic regions induced from the trajectory and map semantics; and $v_e$ exposes the time-varying ego pose for pose-conditioned reasoning. Edges in $\mathcal{E}$ encode object–object and object–place relations derived from world-frame geometry, which can be rendered in the current egocentric frame at query time $t$ via $\mathbf{T}_{e \leftarrow w}(t) = \mathbf{T}_{w \leftarrow e}(t)^{-1}$.

**Object nodes (persistent entities).**   For each instance $k \in \mathcal{I}$, we define the last-seen time $t_k^{\text{last}} = \max \mathcal{T}_k$ and select a keyframe $t_k^{\in \mathcal{T}_k}$ for a single object-level semantic parse (e.g., $t_k^{\text{last}}$ or the highest-confidence detection), avoiding repeated parsing over the full stream. From the keyframe crop, a lightweight parser extracts semantic attributes $\mathbf{s}_k$ (e.g., open-vocabulary label/description and optional attributes). The object geometry is obtained from the world-frame anchors stored in $\mathcal{M}$. To reduce jitter, we optionally compute a stabilized canonical location by temporal pooling:

$$\mathbf{x}_{k,w} = \text{Pool}\big(\{\mathbf{o}_k(t)\}_{t \in \mathcal{T}_k}\big) \in \mathbb{R}^3, \qquad \text{Pool} \in \{\text{median}, \text{mean}\}. \tag{10}$$

We then instantiate an object node $v_k \in \mathcal{V}_o$ that stores

$$v_k : \ \Big(\mathbf{x}_{k,w}, \ \mathbf{s}_k, \ n_k, \ t_k^{\text{last}}, \ \mathcal{K}_k\Big), \tag{11}$$

where $n_k = |\mathcal{T}_k|$ is the observation count and $\mathcal{K}_k \subseteq \mathcal{T}_k$ is a small cache of representative keyframes/timestamps for verification. The tracker identity $k$ is used as the default association key. When identity switches are suspected, we optionally validate associations using a lightweight keyframe-based re-identification score and merge/split nodes accordingly.

**Place nodes (semantic regions).**   We create place nodes by clustering the trajectory into spatial regions with consistent scene semantics. Let $y(t) \in \mathcal{Y}$ be the frame-level scene tag from Step 1 and $\mathbf{p}(t) \in \mathbb{R}^3$ be the ego position from $\mathbf{T}_{w \leftarrow e}(t) = [\mathbf{R}(t) \mid \mathbf{p}(t)]$. For each semantic label $\ell \in \mathcal{Y}$, we collect

$$\mathcal{T}_\ell = \{t \in \{1, \ldots, T\} : y(t) = \ell\}, \tag{12}$$

and cluster the corresponding positions $\{\mathbf{p}(t)\}_{t \in \mathcal{T}_\ell}$ into regions $\{\mathcal{R}_p\}$ (e.g., DBSCAN or k-means in $\mathbb{R}^3$). Each region yields a place node $v_p \in \mathcal{V}_p$ with semantic label $\ell_p \in \mathcal{Y}$ and a compact region descriptor, e.g.,

$$\boldsymbol{\mu}_p = \frac{1}{|\mathcal{R}_p|} \sum_{t \in \mathcal{R}_p} \mathbf{p}(t), \qquad \boldsymbol{\Sigma}_p = \text{Cov}\big(\{\mathbf{p}(t)\}_{t \in \mathcal{R}_p}\big), \tag{13}$$

along with aggregated evidence (timestamps/keyframes) supporting the region. Intuitively, place nodes provide stable anchors for questions referencing semantic regions (e.g., `kitchen`) even when the user has moved away.

**Ego-anchor node.** We include a single ego-anchor node $v_e$ that exposes the evolving pose $\mathbf{T}_{w\leftarrow e}(t)$, enabling pose-conditioned queries. At query time $t$, we use the inverse transform $\mathbf{T}_{e\leftarrow w}(t) = \mathbf{T}_{w\leftarrow e}(t)^{-1}$ to express world-frame entities in the user's current egocentric frame.

**Edges (spatial relations).** Edges $\mathcal{E}$ encode object–object and object–place relations. For object–object relations, we define the world-frame displacement and distance between canonical object locations:

$$\Delta\mathbf{x}_{ij,w} = \mathbf{x}_{j,w} - \mathbf{x}_{i,w}, \qquad d_{ij} = \|\Delta\mathbf{x}_{ij,w}\|_2, \tag{14}$$

and add an edge $(v_i, v_j)$ if $d_{ij}$ is below a threshold or if $j$ is among the $K$ nearest neighbors of $i$. To obtain egocentric directional predicates at query time $t$, we transform object locations into the ego frame:

$$\mathbf{x}_{k,e}(t) = \mathbf{T}_{e\leftarrow w}(t)\,[\mathbf{x}_{k,w}^\top, 1]^\top \in \mathbb{R}^3, \tag{15}$$

and compute the relative vector

$$\Delta\mathbf{x}_{ij,e}(t) = \mathbf{x}_{j,e}(t) - \mathbf{x}_{i,e}(t). \tag{16}$$

Left/right/front/behind relations are then derived from the sign and magnitude of the components of $\Delta\mathbf{x}_{ij,e}(t)$ under the current pose. For object–place relations, we assign object $k$ to place region $\mathcal{R}_p$ if $\mathbf{x}_{k,w}$ falls within the spatial support of that region (or is nearest to it), yielding containment/adjacency edges, optionally with a confidence score.

**Offline build, online inference.** We adopt an offline–online design to decouple memory construction from reasoning. A pose-aware scene graph $\mathcal{G}$ is built offline from the memory buffer $\mathcal{M}$ and cached for efficient inference, such that online reasoning never revisits raw video frames. In addition to the graph, we store spatial captions for each sampled frame, where a VLM describes scene layouts with precise spatial details, including salient objects, relative positions, and inter-object relations. At inference time, given a query timestamp $t$ and the current pose $\mathbf{T}_{w\leftarrow e}(t)$, the model retrieves a relevant subgraph and associated captions, and performs pose-anchored reasoning by projecting world-frame entities into the current egocentric frame via $\mathbf{T}_{e\leftarrow w}(t)$.

### D.3.1. METHOD 2: STREAMING EVALUATION WITH SPATIAL CAPTIONING

**Implementation.** This method reads the raw video file and samples frames at a fixed interval. The system maintains a fixed-size memory buffer of the most recent frames.

**Two-Stage Inference.** In contrast to Method 1, this approach employs a two-stage inference pipeline to improve spatial reasoning capabilities:

1. **Spatial Captioning:** For each sampled frame in the buffer, the VLM generates a detailed spatial caption describing the scene layout. These captions emphasize precise spatial information, including salient objects, their relative positions, inter-object relationships, and overall scene structure. Captions are generated using prompt template 3.

2. **Visual-Textual Question Answering:** The model receives an interleaved sequence of frames and their corresponding spatial captions as input. This design explicitly grounds the reasoning process in both visual evidence and structured spatial descriptions. We use prompt template 4 for this stage.

**Hyperparameters.** We set the buffer size, and image resolution as hyperparameters. We maintain a buffer of 64 frames, and resize images to 512×512 pixels.

### D.4. Detailed Implementation Parameters

All experiments are conducted on a server equipped with $8\times$ NVIDIA RTX A6000 GPUs (48GB each) and $4\times$ NVIDIA A100 GPUs (80GB each). For all models, we perform inference using HuggingFace's Transformers library, with model-specific adaptations following their official implementations. To ensure reproducibility and consistency, each model is evaluated within its own isolated virtual environment, strictly adhering to the corresponding official dependencies and

configurations. For video inputs, all frames are resized such that the longer side is at most 512 pixels while preserving the original aspect ratio. Unless otherwise specified, we use deterministic decoding with top-$p = 1.0$ and temperature $= 0.0$ across all experiments.

## D.5. Prompts

**Prompt 1: Uniform Sampling Prompt**

```
Role:  user
Content:  [
  // Input:  Uniformly sampled frames
  { "type":  "image", "image":  <Frame 1> },
  { "type":  "image", "image":  <Frame 2> },
  ...
  { "type":  "image", "image":  <Frame K> },
  // System Instruction
  { "type":  "text", "text":
    "You are an egocentric video QA assistant.
    This is a multiple-choice question answering task based strictly on visual
content.
    You will see a sequence of frames sampled uniformly from the start of the video
    up to the question time <T> seconds.
    Use only the frames as evidence to answer the question.

    Rules:
    - Rely only on visible evidence.
    - Do not hallucinate unseen objects or actions.
    - Use temporal reasoning if helpful.
    - There is exactly one best answer.
    - Only output a single uppercase letter (A, B, C, ...).
    - Do not output any explanations or extra text.

    Your job is to choose the option that best matches the visual content.

    Question:
    <Question Text>

    Options:
    A. <Option A>
    B. <Option B>
    C. <Option C>
    ...

    There are <K> frames in temporal order.
    Only output the letter of the best option (A, B, C, ...).
    Note that there is only one best option.
    Output format:  answer is :..."
  }
]
```

**Prompt 2: Stream Sampling Prompt**

```
Role: user
Content: [
    // Input: Stream-selected frames from recent time window
    { "type": "image", "image": <Frame 1> },
    { "type": "image", "image": <Frame 2> },
    ...
    { "type": "image", "image": <Frame K> },
    // System Instruction
    { "type": "text", "text":
      "You are an egocentric video QA assistant.
      This is a multiple-choice question answering task based strictly on visual
content.
      You will see a sequence of frames selected by stream within the last <K> seconds
      up to the question time <T> seconds.
      Use only the frames as evidence to answer the question.

      Rules:
      - Rely only on visible evidence.
      - Do not hallucinate unseen objects or actions.
      - Use temporal reasoning if helpful.
      - There is exactly one best answer.
      - Only output a single uppercase letter (A, B, C, ...).
      - Do not output any explanations or extra text.

      Your job is to choose the option that best matches the visual content.

      Question:
      <Question Text>

      Options:
      A. <Option A>
      B. <Option B>
      C. <Option C>
      ...

      There are <K> frames in temporal order.
      Only output the letter of the best option (A, B, C, ...).
      Note that there is only one best option.
      Output format:  answer is :..."
    }
]
```

**Prompt 3: Spatial Caption Generation Prompt**

```
Role:  user
Content:  [
    // Input:  Single frame to be captioned
    { "type":  "image", "image":  <Frame> },
    // Caption Generation Instruction
    { "type":  "text", "text":
      "You are a vision expert for egocentric frames.
      Describe the scene with a focus on precise spatial layout.

      Include:
      - All salient objects and people (with attributes if visible)
      - Relative positions (left/right, front/behind, above/below, near/far)
      - Object-to-object relations (on, under, inside, next to, touching)
      - Room/scene structure and landmarks (walls, doors, counters, shelves, floor)
      - Actions/interactions only if clearly visible

      Write 3-6 concise sentences.  Be strictly factual.  Do not guess."
    }
]
```

## Prompt 4: QA with Spatial Captions Prompt

```
Role: user
Content: [
  // Input: Retrieved frames with captions
  { "type": "text", "text": "[Frame 1/<K> | t=<T_1>s]" },
  { "type": "image", "image": <Frame 1> },
  { "type": "text", "text": "Spatial caption: <Caption 1>" },
  { "type": "text", "text": "[Frame 2/<K> | t=<T_2>s]" },
  { "type": "image", "image": <Frame 2> },
  { "type": "text", "text": "Spatial caption: <Caption 2>" },
  ...
  { "type": "text", "text": "[Frame <K>/<K> | t=<T_K>s]" },
  { "type": "image", "image": <Frame K> },
  { "type": "text", "text": "Spatial caption: <Caption K>" },
  // Final QA Instruction
  { "type": "text", "text":
    "You are an egocentric video QA assistant.
    You are given a set of frames (in temporal order) from the start of the video
    up to the question time.
    Each frame is accompanied by a spatial caption. Use ONLY the provided frames
    and captions as evidence.

    Rules:
    - Rely only on visible evidence from the frames and captions.
    - Do not hallucinate unseen objects, actions, or events.
    - Use temporal reasoning if helpful.
    - There is exactly ONE best answer.
    - Output ONLY a single uppercase letter (A, B, C, ...). No explanations.

    Question time: <T_question> seconds
    Number of selected frames: <K>

    Question:
    <Question Text>

    Options:
    A. <Option A>
    B. <Option B>
    C. <Option C>
    ...

    Answer (single letter only):"
  }
]
```

## Prompt 5: QA with Spatial Captions + Scene-Graph Summary Prompt

```
Role: user
Content: [
  // Frames with spatial captions (temporal order)
  { "type": "text", "text": "[Frame 1/<K> | t=<T_1>s]" },
  { "type": "image", "image": <Frame 1> },
  { "type": "text", "text": "Caption: <Caption 1>" },
  ...
  { "type": "text", "text": "[Frame <K>/<K> | t=<T_K>s]" },
  { "type": "image", "image": <Frame K> },
  { "type": "text", "text": "Caption: <Caption K>" },

  // Pose-anchored scene-graph summary at question time
  { "type": "text", "text": "[Graph Summary @ t=<T_question>s]" },
  { "type": "text", "text":
    "Use the following structured summary rendered in the current egocentric frame:
    Orientation: front/left/right/rear (relative to the wearer), with optional
distance near/far.
    Objects: <Object List>
    Relations: <Pose-anchored Relations>

    Object List format (one per line):
    obj=<name> | tag=<semantic_tag> | color=<color> | count=<n> |
traj=<brief_trajectory> | last=<t>s
    Relations format (one per line):
    <objA> is <orientation> (<near/far>) of you
    <objA> is near/in/on <objB> "
  },

  // Final QA instruction
  { "type": "text", "text":
    "You are an egocentric video QA assistant. Use ONLY the provided frames,
captions, and graph summary.
    Choose exactly ONE best option. Output ONLY a single uppercase letter (A, B, C,
...).

    Question time: <T_question>s
    Frames: <K>

    Question: <Question Text>
    Options: A. <A>  B.   C. <C>  ..."
  }
]
```

## Prompt 6: Generate Binary Distractor Prompt

**Role:** system
**Content:** [
   "You are an expert test-item writer specializing in BINARY (two-choice)
questions.
   Your job is to write ONE plausible but WRONG alternative that would be the other
choice.
   The wrong option must be mutually exclusive with the true answer, and should look
like
   a natural answer to the question."
]

**Role:** user
**Content:** [
   "Question: {question}
   True Answer: {gold}
   {existing_section}
   Write EXACTLY ONE wrong option for a binary (two-choice) question.

   CORE RULE (BINARY FLIP):
   - Use the question to identify the two alternatives (often connected by 'or').
   - Output the OTHER alternative, not the one selected by the true answer.

   COMMON Examples:
   - left <-> right
   - yes, you can <-> no, you can't.
   - Dumbbells are closer to you.  <-> Dumbbells are further away from you.

   STYLE & STRUCTURE:
   - Match the true answer's tone, sentence pattern, and grammatical structure
   - Keep the same key subject/entities; only switch the binary choice
   - Keep the same perspective and tense as the true answer
   - Keep length similar to the true answer

   CONSTRAINTS:
   - English only, <= 18 words, end with a period
   - No hedging ('probably', 'not sure', 'insufficient information')
   - No 'All/None of the above'
   - Must be different from all existing wrong options

   Output:  ONE line only.  No numbering.  No quotes."
]

## Prompt 7: Generate multiple distractors Prompt

**Role:** system
**Content:** [
    You are an expert test-item writer.  Given a question, the TRUE
    answer, and possibly several seed near-miss candidates, generate
    EXACTLY <n_lines> plausible but WRONG options.

    Each option must be:
    1) English-only, concise (<= 18 words), ending with a period.
    2) Mutually exclusive with the true answer (not a paraphrase; not
       entailed).
    3) Highly similar in structure/content (near-miss):  change a concrete
       attribute like left/right, count, object name, container/surface,
       path step, floor level, or order.
    4) Avoid generic/hedging phrases (e.g., 'Insufficient information',
       'Probably', 'Not sure').
    5) Avoid 'All/None of the above'.
    6) Keep the same answer TYPE/slot (location/quantity/trajectory
       etc.).
    7) If seeds are provided, use them as inspiration but improve and
       diversify them.  If seeds are few or absent, CREATE new
       high-quality distractors from scratch.
    8) MATCH THE STYLE AND STRUCTURE of the true answer closely:
        – Use similar sentence patterns, phrasing, and grammatical
          structure.
        – If the question mentions a subject (e.g., 'apple') and the
          true answer also includes it, ALL options must include that
          same subject to maintain consistency.
        – Mirror the level of detail and specificity in the true answer.
        – Keep the same perspective (first/third person) and tense as
          the true answer.
    9) Generate diverse options covering different types of plausible
       errors (spatial, temporal, quantitative, categorical, etc.)  while
       maintaining the near-miss quality.
]

**Role:** user
**Content:** [
    "Question:  {question}
    True Answer:  {gold}
    seed_section
    IMPORTANT: Analyze the style of the true answer carefully.  Generate
    options that:
    – Follow the exact same answering pattern and structure as the true
      answer
    – Include the same key subjects/entities mentioned in both question
      and true answer
    – Maintain stylistic consistency while being factually incorrect
    – Even if seeds are limited or missing, you MUST generate ALL
      n_lines unique options
    – Ensure variety:  don't just change one attribute; explore different
      plausible mistakes

    Write EXACTLY n_lines lines, one option per line, no numbering,
    end each with a period."
]
*Note:* seed_section is conditionally inserted as:
    "Seed candidates (use as inspiration, but create variations):
    – seed_1
    – seed_2
    ..."  (if seeds are provided)
or:
    "No seed candidates provided.  Create distractors from scratch."

## Prompt 8: Generate distractors for Multi-Choice Prompt

```
Role: system
Content: [
   "You are an expert test-item writer.  Generate plausible but incorrect
   options for multiple-choice questions.  Focus on creating near-miss
   distractors that are structurally similar to the correct answer but
   factually wrong."
]
Role: user
Content: [
   "Question:  question
   True Answer:  gold
   existing_section
   Generate EXACTLY ONE new wrong option that:

   STYLE & STRUCTURE:
   - Match the true answer's sentence pattern, phrasing, and grammatical
     structure
   - Include the same key subjects/entities from the question and true
     answer
   - Mirror the level of detail and specificity
   - Use the same perspective and tense

   CONTENT REQUIREMENTS:
   - Change ONE concrete attribute:  left/right, count, object name,
     location, order, etc.
   - Be mutually exclusive with the true answer (not a paraphrase)
   - Be distinctly different from all existing options
   - Explore a different type of error than existing options
     (spatial/temporal/quantitative/categorical)

   CONSTRAINTS:
   - English only, ≤ 18 words, end with a period
   - No hedging phrases ('probably', 'not sure', 'insufficient
     information')
   - No 'All/None of the above'

   Output:  Write ONE line only, no numbering or formatting."
]
Note:  existing_section is conditionally inserted as:
   "Existing wrong options (your new option MUST be different):
   - option_1
   - option_2
   ..."  (if existing options are present)
```

## Prompt 9: Fix Entailment Prompt

**Role:** user

**Content:** [

You are helping to fix multiple-choice question options that have entailment issues.

Problem: The following options show entailment relationship (one option logically implies another), which makes the question invalid.

Question: {question}

Current Options: {formatted_options}

Correct Answer: {correct_label}. {correct_answer}

Problematic Options:

– Option {label_a}: "{option_i}"

– Option {label_b}: "{option_j}"

(max_entailment score: {max_entailment})

Root Cause: These options likely differ only in {difference_type},

which creates a logical entailment relationship.

Your Task: Generate new replacement option(s) for the problematic option(s): {problematic_labels}

CRITICAL REQUIREMENTS:

1. Change CORE attributes instead of adding/removing positional information:

  – Change colors

  – Change objects

  – Change quantities

  – Change states

2. Maintain identical sentence structure as the correct answer

3. Keep length consistent (±2 words from the correct answer length)

4. Ensure mutual exclusivity with every existing option

5. Keep it plausible but clearly wrong for the question

Output Format (JSON only):

{json_template}

]

## Prompt 10: Fix Length Imbalance Prompt

**Role:** user

**Content:** [

   You are helping to fix multiple-choice question options that have length imbalance issues.

   Problem: Some options are significantly longer or shorter than the correct answer, making it easy to guess.

   Question: {question}

   Current Options:     {formatted_options_with_length}

   Correct Answer: {correct_label}. {correct_answer} ({target_length} words)

   Your Task: Rewrite the problematic options to match the target length

   ({target_length} ± 2 words): {problematic_labels}

   CRITICAL REQUIREMENTS:

   1. Match the length: each rewritten option should be {target_length} ± 2 words

   2. Keep the same grammatical structure as the correct answer

   3. Maintain content diversity (no meaningless padding)

   4. Preserve wrongness and avoid entailment with other options

   Output Format (JSON only):

   {json_template}

]

## Prompt 11: Blind Test Prompt for Egocentric Video QA

```
Role:  system
Content:  [
    "You are a helpful assistant designed to output JSON."
]
Role:  user
Content:  [
    "You are an egocentric video QA assistant taking a 'blind test'.
    This implies you DO NOT have access to the visual content/video
    frames, but you must guess the most likely answer based on language
    bias, common sense, or the logical relationships between the question
    and options.

    Rules:
    - Analyze the question and options purely textually.
    - Select the best option.
    - **Explain WHY you chose this option without seeing the video.**
      (e.g., 'Option A is the most common action associated with
      this object', or 'Option B is the only grammatically correct
      answer', etc.)
    - Output your response in strictly Valid JSON format.

    Question:
    {current_q}

    Options:
    {options_text}

    Video context info:  There are {num_frames} frames (which you cannot
    see).

    Required JSON Output Format:
    {
      "reason":  "Your concise reasoning here...",
      "answer":  "The single uppercase letter (A, B, C...)"
    }"
]
Note:  {current_q}, {options_text}, and {num_frames} are dynamically
    populated from the current question being evaluated.
```

## Prompt 12: Generate distractors Prompt

```
Role:  user
Content:  [
    "Please generate 5 deceptive incorrect options for the following question.
    The options should appear reasonable but are actually incorrect.
    Utilize information gaps to create mutual corroboration among the
    confusing options, thereby leading the model to select one of them.
    Create at least one longer option whose information overlaps and
    intersects with other options.  Preferably include 'you' in the options.
    Respond in the tone of an assistant.

    Question:  {question}
    Correct Answer:  {answer}

    Please output the options directly, one per line."
]
```

# E. Additional Experiments

## E.1. Hyperparameter Comparisons

In this section, we provide a detailed analysis of the impact of key hyperparameters on model performance, including model size, spatial resolution, and temporal context length. All experiments in this subsection are conducted using the Qwen series as the backbone.

### E.1.1. IMPACT OF SPATIAL RESOLUTION

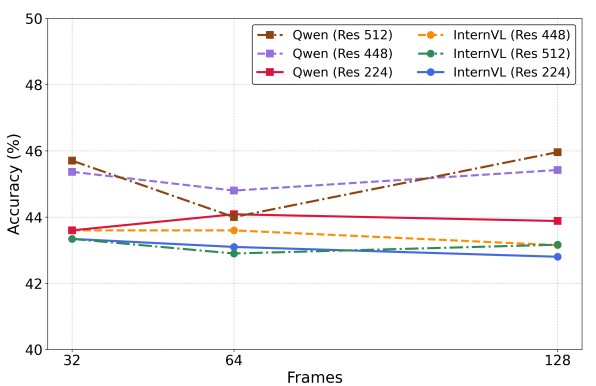 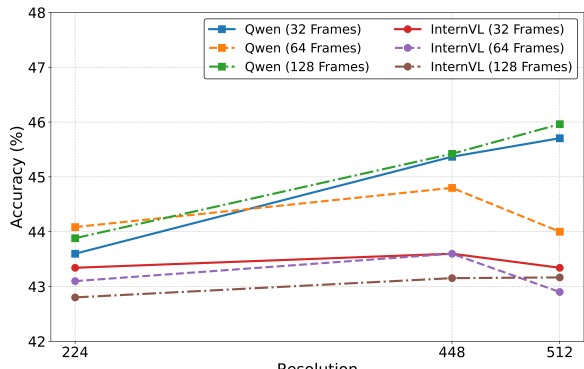

*(a)* **Temporal Context:** The impact of input frame counts (32, 64, 128) on the 8B model's accuracy.

*(b)* **Spatial Resolution:** The effect of varying input image resolutions (224, 448, 512) on the 8B model.

*Figure 9.* Effect of temporal and spatial input parameters on model accuracy: A study of Qwen3-VL-8B-Instruct and InternVL3.5-7aB-Instruct.

Figure 9a shows that temporal context affects Qwen and InternVL differently. Qwen exhibits a clear resolution-dependent temporal pattern: at low resolution (Res 224), accuracy peaks at 64 frames (44.08%) and then slightly declines, suggesting that longer temporal sequences may introduce redundancy when spatial information is limited. At high resolution (Res 512), Qwen shows a non-monotonic trend, dropping at 64 frames but reaching the best result at 128 frames (45.9%), indicating that sufficient spatial detail better supports long-range temporal reasoning. In contrast, InternVL remains relatively stable across different frame counts, with only minor fluctuations, suggesting stronger temporal robustness but limited benefit from extended temporal context.

Figure 9b further investigates the impact of spatial resolution. Qwen benefits more clearly from increased resolution, especially under the 32-frame and 128-frame settings, where accuracy generally improves from Res 224 to Res 512. However, the 64-frame setting peaks at Res 448 and decreases at Res 512, indicating that higher spatial resolution is not always beneficial under all temporal configurations. By comparison, InternVL is less sensitive to spatial resolution, with accuracy remaining stable around 43% across different resolutions.

**Cross-Model Comparison.** In terms of peak performance, Qwen3-VL-8B-Instruct achieves the highest accuracy, reaching 45.96% at Res 512 with 128 frames, surpassing InternVL3.5-7B-Instruct by roughly 2.36 percentage points. However, this advantage comes at the cost of configuration sensitivity, as Qwen exhibits larger performance variations across temporal and spatial settings, whereas InternVL maintains more stable but generally lower accuracy. From an efficiency perspective, Qwen configured at Res 448 with either 32 or 128 frames offers a favorable trade-off, achieving accuracy close to the best setting while reducing computational cost compared with Res 512.

**Application Implications.** These findings suggest that model selection and configuration should be guided by application characteristics. Qwen3-VL-8B-Instruct with Res 512 and 128 frames is better suited for scenarios requiring fine spatial details and long-range temporal understanding, such as continuous object tracking or long-horizon action analysis. In contrast, Qwen at Res 448 with 32 or 128 frames provides a more efficient alternative when computational resources are limited, while still preserving competitive accuracy. InternVL3.5-7B-Instruct, although less accurate overall, may be preferable for fragmented or variable-quality user-generated videos where robustness to input variation is more important than peak performance.

E.1.2. IMPACT OF TIME INTERVALS.

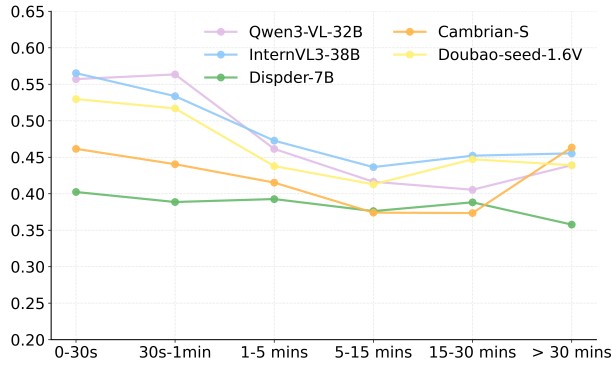

*Figure 10.* Accuracy trends across Evidence Span.

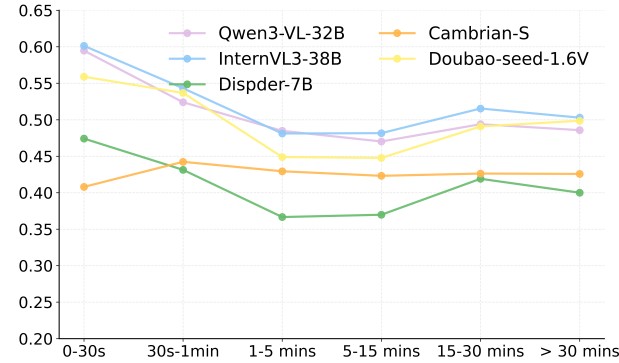

*Figure 11.* Accuracy trends across different time intervals from video start to question timestamp.

Based interval analysis, we observe a consistent accuracy degradation pattern across all models in figure 10, with performance declining from short intervals (0-30s) to medium intervals (1-15 mins). The steepest drop occurs in the 1-5 minute window, where high-performing models like Qwen3-VL-32B and InternVL3-38B experience 6-10 percentage point decreases. Notably, some models show partial recovery at intervals exceeding 30 minutes, suggesting different reasoning mechanisms for very long temporal distances. Higher-capacity models demonstrate better absolute performance but suffer larger relative degradation (22-27%), while smaller models like Dispder-7B show more stable but consistently lower accuracy (11% relative drop). These findings indicate that current vision-language models struggle most with medium-range temporal reasoning, highlighting a critical limitation in their temporal context windows that warrants further architectural improvements.

As illustrated in Figure 11, most models exhibit a notable decline in accuracy as the time interval between video start and question timestamp increases, suggesting temporal forgetting effects in long-form video understanding. Qwen3-VL-32B demonstrates the most stable performance, maintaining relatively high accuracy (above 0.55) even at intervals exceeding 30 minutes. In contrast, smaller models like Dispder-7B show more pronounced degradation, dropping from approximately 0.50 in the 0-30s interval to around 0.40 beyond 30 minutes. Interestingly, all models demonstrate relatively consistent performance in the middle intervals (5-15 mins and 15-30 mins), with accuracy variations primarily occurring at the earliest (0-30s) and latest (>30 mins) time windows.

Robustness of SOTA Models: Leading models such as InternVL3-38B and Qwen3-VL-32B demonstrate remarkable stability, with average scores fluctuating by less than 0.5 points between strategies. This suggests that large-scale models possess robust long-context capabilities that are not heavily biased by the distribution of input frames.

Task-Specific Impacts:

- **Object Memory:** Uniform sampling generally benefits global retrieval tasks. For instance, Qwen2.5-VL-7B sees an improvement in the *Category & Quantity* category (Avg. increases from 42.5 with stream sampling to 45.5 with uniform sampling), as uniform frames provide better coverage of objects appearing earlier in the episode.

- **Spatial State Estimation:** Conversely, tasks requiring fine-grained motion continuity, such as *Ego-centric Position*, favor recent frame sampling. Notable sensitivity is observed in VG-LLM-8B, where performance drops significantly (Avg. $38.34 \rightarrow 32.90$) under uniform sampling, likely due to the loss of immediate temporal continuity required for its spatial reasoning mechanism.

Streaming Models: As expected, streaming models (e.g., Dispder-7B, Flash-VStream) show almost identical performance across settings, validating their internal memory mechanisms' independence from offline sampling heuristics.

### E.2. Spatial Captions

To enhance the model's understanding of spatial relationships in the video frames, we employ an alternative approach that augments visual inputs with spatial descriptions. After selecting the 64 most recent frames relative to the question timestamp,

we use a VLM to generate a detailed spatial caption for each individual frame. These captions specifically describe the spatial relationships present in each image, such as object positions, orientations, and relative arrangements.

The generated spatial captions are then concatenated with their corresponding frames and fed together into the VLM for final answer inference. This approach aims to provide explicit spatial reasoning signals that complement the visual information, potentially improving the model's ability to answer spatially-grounded questions.

The prompts used for this method are provided in Prompt 3 for caption extraction and Prompt 4 for the question-answering stage.

## F. Dataset and Annotations

### F.1. Dataset Source

We utilize several publicly available datasets, each contributing specific subsets tailored for spatial reasoning, temporal perception, and embodied assistance tasks.

EgoLife (Yang et al., 2025a): We select samples from EgoLife to evaluate the understanding of continuous daily life activities and social interactions. As a dataset capturing multi-view egocentric footage of participants living together over extended periods (up to a week), it allows us to test the model's ability to maintain long-term consistency and interpret complex interpersonal dynamics within a shared 3D environment.

HourVideo (Chandrasegaran et al., 2024): To challenge models with extremely long-context reasoning, we incorporate samples from the HourVideo dataset. With videos ranging from 20 to 120 minutes, this subset serves as a rigorous test for the model's capacity to retrieve and synthesize information across extensive temporal windows, evaluating its episodic memory and ability to reason about events that span significant timeframes.

ScanNet (Dai et al., 2017): We utilize indoor scene sequences from ScanNet to focus specifically on 3D spatial perception and scene geometry. By leveraging its rich RGB-D data and reconstructed 3D meshes of complex indoor environments, this subset assesses the model's grasp of static spatial relationships, object placement, and volumetric understanding, which are critical for embodied navigation tasks.

EPIC-KITCHENS (Damen et al., 2020): We incorporate this dataset to target fine-grained action recognition and detailed object interaction. Capturing unscripted daily activities in kitchen environments, this subset challenges models to identify rapid hand-object manipulations and subtle state changes (e.g., chopping, washing). It serves as a critical test for the model's precision in perceiving short-term temporal dynamics and procedural understanding within object-dense settings.

EgoBlind (Xiao et al., 2025): We include the EgoBlind dataset to introduce challenges related to assistive AI for the visually impaired. This subset features egocentric video captured by blind individuals, often containing unique visual constraints such as unstable motion or off-center framing. It tests the model's robustness and its ability to perform safety-critical reasoning (e.g., obstacle detection) and navigation assistance under suboptimal visual conditions.

TeleEgo (Yan et al., 2025): We introduce the TeleEgo dataset, leveraging its long-duration streaming video characteristics to evaluate models' continuous perception capabilities for dynamic spatial relationships. Through analyzing environmental evolution from an egocentric perspective, the dataset assesses models' robustness in capturing instantaneous spatial alignment and action causality within complex 3D scenes.

### F.2. Dataset Statistics

#### F.2.1. QUESTION TEMPLATE

As illustrated in Table 6, we distill the natural language queries generated by human annotators—guided by our dynamic spatiotemporal protocols—into four representative categories. This taxonomy reflects the core dimensions prioritized by annotators during the interpretation of streaming videos. Statistical analysis of the annotation corpus reveals a significant emphasis on ego-centric spatial awareness and trajectory tracking, necessitating that models comprehend the continuous relative evolution between the observer and the environment. Furthermore, interrogative patterns concerning proximity, reachability, and object quantity variations indicate that human observers engage in deep semantic reasoning regarding physical geometric attributes and event-driven state transitions within dynamic scenes.

In conclusion, these templates derived from manual annotations not only preserve the inherent diversity of natural expression

*Table 6.* Question Templates for tasks in UCS-Bench. We replace the highlighted part in the question template from scene to scene to construct our benchmark. Note that a complete example question is provided for Route Plan.

| Task | Question Template |
|---|---|
| **Ego-centric Position & Orientation Memory** | · **Ego-centric Position:** *When I first entered the {location}, in what direction was the {object} relative to my orientation?*
· **Relative Position & Orientation Memory:** *Where in the {location} did I place the {object} after completing the {action}?* |
| **Embodied trajectory & Movement Memory** | · **My Trajectory:** *What path did I take while moving toward the {object}?*
· **Object Trajectory:** *What was the movement trajectory of the {object} as it moved from {location} to {location}?* |
| **Perceived Proximity & Reachability Memory** | · **Distance Comparison Memory:** *What is the direct distance between {object} and {object} within the {location}?*
· **Reachability Judgment:** *When I was positioned at the {location}, was I able to directly reach the {object}?* |
| **Object Recall & Quantity Tracking** | · **Object Category Recall:** *Before the {event} occurred, how many types of {category} were present at the {location}?*
· **Quantity Change Tracking:** *How many {category}(s) were added or removed during the {event}?* |

but also precisely encompass hierarchical challenges ranging from geometric localization to spatiotemporal logical reasoning. This effectively underscores the benchmark's specificity and comprehensiveness in evaluating dynamic variations in spatial relationships.

### F.2.2. WORD CLOUD

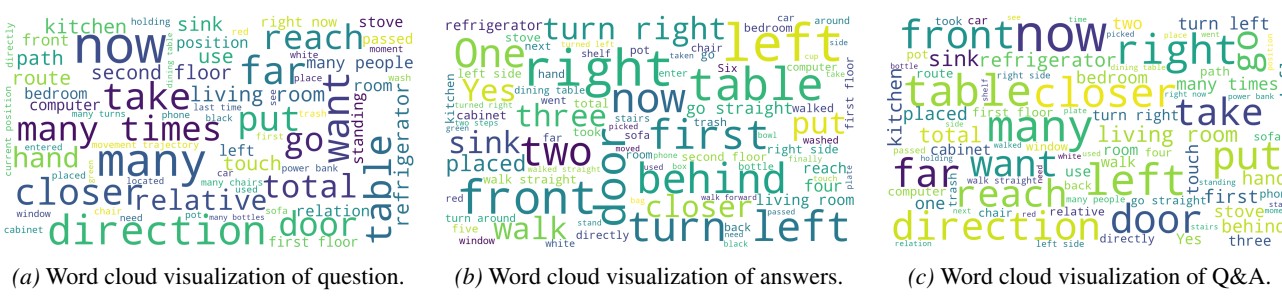

*(a)* Word cloud visualization of question.  *(b)* Word cloud visualization of answers.  *(c)* Word cloud visualization of Q&A.

*Figure 12.* Word cloud visualization of UCS-Bench

The word cloud 12a of the questions reveals the critical dimensions required for evaluating multimodal models in processing streaming videos. Prominent keywords such as "direction", "relative", "closer", and "far" occupy central positions, indicating that questions in the dataset place strong emphasis on dynamic spatial localization and relative distance perception. The frequent appearance of temporal markers such as "many times" and "now" indicates that models need to possess long-term memory capabilities to track action frequencies and instantaneous state changes in streaming videos.

The answer word cloud 12b shows a complementary pattern where directional terms ("right," "left," "front"), sequential indicators ("first," "second," "one," "two"), and spatial descriptors ("behind," "closer," "table") are most frequent, demonstrating that responses require precise spatial localization and tracking of object positions as they evolve through the video stream.

The word cloud 12c reveals a distinct focus on egocentric interaction and dynamic navigation, where directional terms ("front", "right", "left", "direction", "behind"), spatial-temporal descriptors ("closer", "far", "now"), and interaction verbs ("take", "put", "reach", "walk") are most predominant. This pattern demonstrates that the queries necessitate precise user-centric spatial perception, requiring the model to reason about dynamic scenes as they evolve through the video stream ("now") and track fine-grained object interactions and relative positions within indoor environments ("kitchen", "living room", "table").

## F.3. Annotation System

### F.3.1. ANNOTATION INTERFACE AND WORKFLOW

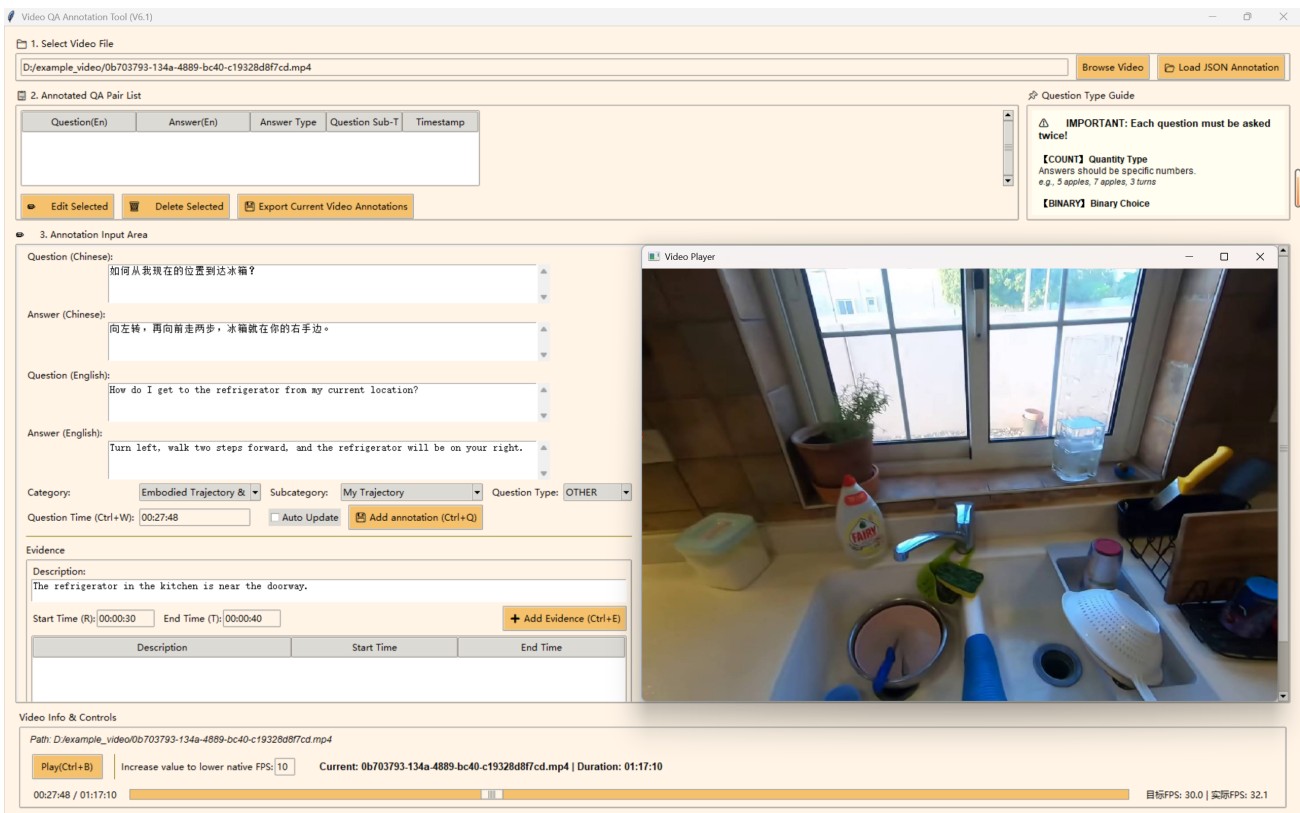

*Figure 13.* The video QA annotation tool. This tool enables annotators to view egocentric video streams and curate question-answer pairs with precise temporal grounding and spatial evidence.

To facilitate high-quality data collection, we developed a specialized annotation tool, whose UI is shown in Figure 13. It provides structured fields for annotators to input question-answer pairs, question timestamps, evidence timestamps, question categories, and question types. Additionally, built-in temporal logic constraints prevent incorrect labeling of question and evidence timestamps. For example, the 'question timestamp' is constrained to occur strictly after the 'evidence frame', ensuring that the dataset evaluates genuine temporal reasoning.

The interaction is designed to be highly efficient. Shortcuts (e.g., `Ctrl+S` for saving, `Ctrl+T` for perspective switching) streamline the manual labor. The system supports batch processing for algorithmic tasks (e.g., batch optimization for counting questions) while enforcing human verification for semantic nuances.

### F.3.2. DATA REFINEMENT AND VERIFICATION PROCESS

To ensure the high quality of our dataset and mitigate language biases, we developed a specialized QA refinement and verification tool, as shown in Figure 14. The tool facilitates an iterative refinement process combining rule-based heuristics, LLM assistance, and human verification. The refinement tool consists of four primary modules:

**Metadata Panel:** Displays video metadata, current question-answer pairs, and automated quality assessment reports.

**Editor Panel:** Allows real-time modification of questions, answer choices, and question types.

**AI Assistant Module:** Integrated with GPT-4o, this module aids in generating challenging distractors and performing logic-based optimizations for counting-type questions.

**Adversarial Validation:** We employ a separate LLM agent to attempt answering questions without visual input, ensuring that options do not leak information that enables text-only reasoning. Integrated with a Large Language Model (e.g.,

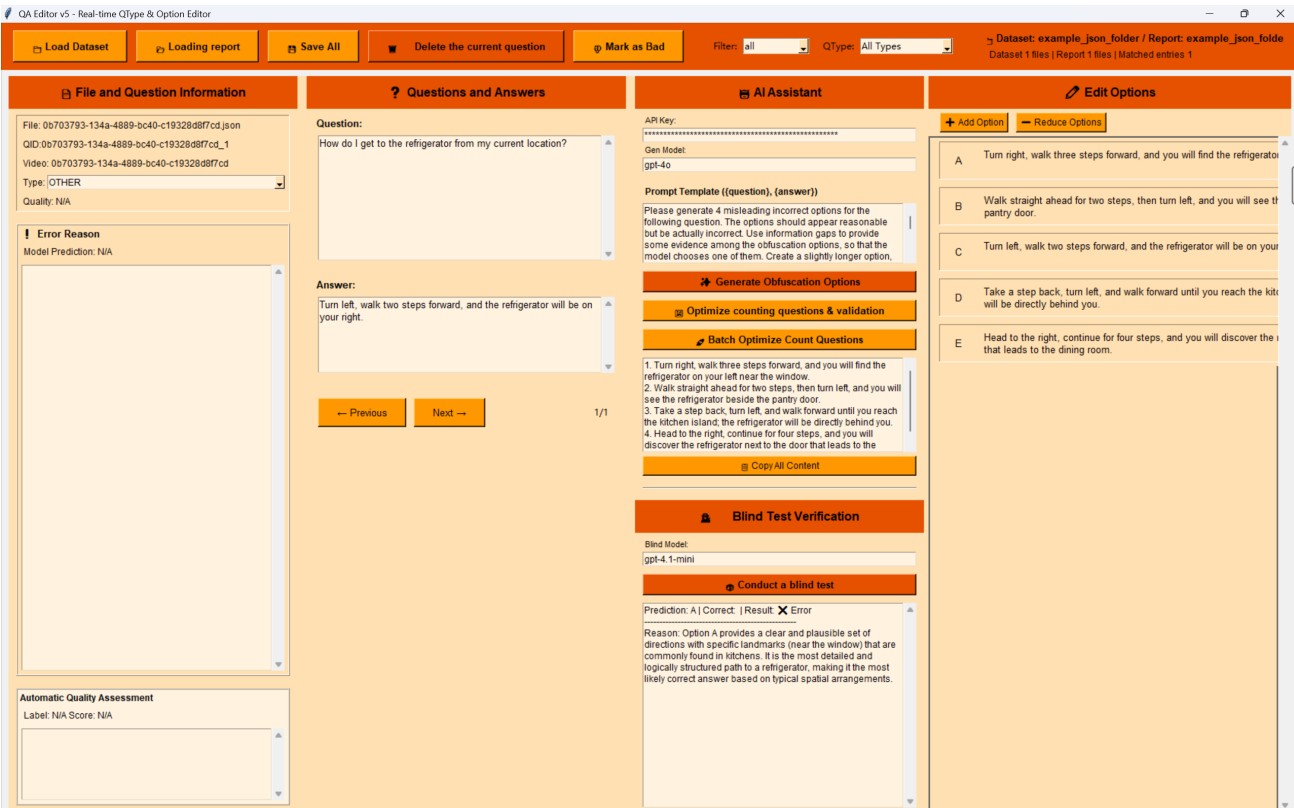

*Figure 14.* The QA refinement and verification tool. This interface leverages Large Language Models to automatically generate challenging distractors and conduct blind-test verification to ensure the quality of the multiple-choice dataset.

GPT-4o), this module assists experts by generating "plausible but incorrect" distractors based on specific prompts.

The data optimization process follows a sequential workflow for each data sample $x_i$:

**Step 1: Initialization and Pre-filtering** The system loads the raw dataset along with pre-computed quality reports. Annotators utilize the filter to prioritize samples marked as 'bad' or specific question types (e.g., *Count*) that require intensive correction.

**Step 2: Textual Normalization and Perspective Correction** Since many raw captions are egocentric, the system implements a rule-based regex module to automatically convert first-person perspectives (e.g., 'I', 'my') to second-person perspectives (e.g., 'you', 'your'). This ensures the question aligns with the user's perspective in a VQA setting.

**Step 3: Option Optimization** We apply distinct strategies based on the Question Type:

- **General Questions:** The annotator invokes the AI Assistant with a prompt designed to generate five plausible but incorrect distractors. The prompt enforces information overlap among options to increase difficulty and reduce elimination shortcuts.
- **Counting Questions:** A specialized heuristic algorithm is triggered to handle numerical counting queries. The system first parses the numerical value $N$ from the ground truth answer. It then automatically generates a set of candidate options, such as $\{N, N+1, N+2, N+3, N+4\}$, or alternative distributions where $N$ is not positioned at the median of the option set. And it ensures that the options are strictly numerical or consistently textual (e.g., "three" vs. "3").

**Step 4: Adversarial Blind Test Loop** Before saving, the annotator executes the *Blind Test*.

- The system sends the text-only tuple (Question, Options) to an evaluator model. The evaluator model attempts to guess the answer based on language bias or common sense, without access to video frames.

- If the blind evaluator correctly predicts the answer, the question is flagged as biased. Then, the annotator must then modify the question phrasing or the options to remove the textual leakage and re-run the blind test until the evaluator fails or provides a random guess.

**Step 5: Final Quality Decision** Upon passing the blind test, the annotator assigns a final quality label. Only the validated samples are saved to the persistent storage.

### F.4. Generation Distractor Pipeline

#### F.4.1. QUESTION TYPE

To systematically evaluate models' capability in understanding dynamic spatial relationships from a user-centric perspective, we design a hierarchical question taxonomy that mirrors the cognitive complexity of real-world spatial reasoning tasks. Specifically, we categorize questions into three types based on the nature of user's spatial cognition needs: binary, count, and multi-choice.

**Binary questions** capture the most fundamental level of spatial understanding—making definitive spatial judgments. These questions reflect scenarios where users need immediate, actionable spatial awareness, such as "Can I reach the fridge now?" or "Was the fridge on my left or right?". This question type evaluates whether models can provide clear, binary spatial assessments that directly support user decision-making.

**Count questions** target quantitative spatial perception, requiring models to track and enumerate objects in dynamic environments. Questions like "How many objects are within reach?" assess models' ability to maintain precise spatial awareness over time—a critical capability for users navigating changing environments.

**Multi-choice questions** encompass more complex spatial reasoning scenarios that cannot be reduced to simple binary or numerical answers. These questions evaluate models' capacity to handle nuanced, context-dependent spatial relationships that users frequently encounter in real-world interactions.

#### F.4.2. MULTI-STAGE DISTRACTOR PIPELINE

To rigorously assess models' spatial understanding capabilities, we design a distractor generation strategy that mirrors the cognitive challenges users face when interpreting dynamic spatial relationships. The number and nature of distractors vary by question type to create realistic confusion scenarios: binary questions include one distractor, while count and multi-choice questions feature four distractors each.

**Binary questions.** For binary spatial judgments, we generate distractors that represent the most plausible alternative interpretation. Specifically, we construct one distractor that presents the opposite conclusion to the ground-truth answer. This design directly tests whether models can make definitive spatial decisions rather than hedging between competing interpretations—a critical capability for supporting user actions in dynamic environments. The distractor is generated using LLM with Prompt 6.

**Count questions.** Quantitative spatial errors often manifest as off-by-one or magnitude estimation failures in human cognition. To simulate these realistic error patterns, we replace the correct count in the ground-truth answer with erroneous numbers. Crucially, we strategically select these numbers such that the correct value becomes the second largest or second smallest among the options. This design prevents trivial elimination strategies (e.g., always choosing the median) and forces models to perform genuine spatial enumeration rather than statistical guessing.

**Multi-choice questions.** For complex spatial reasoning questions, we employ a sophisticated five-stage pipeline that generates semantically plausible yet incorrect distractors—mirroring the nuanced misunderstandings users might have about spatial relationships in dynamic environments. This multi-stage approach ensures that our evaluation captures authentic spatial reasoning challenges rather than superficial linguistic patterns. Given a source question $q$ and its ground-truth answer $a^*$, our pipeline operates as follows:

**Stage 1: Neighbor-Based Candidate Retrieval**

To generate distractors that reflect realistic user confusion patterns, we retrieve answers from semantically similar spatial reasoning questions built in the same data source. This retrieval-based approach ensures that our distractors mirror genuine spatial understanding errors observed in real question-answer scenarios.

Specifically, we encode question-answer pairs as:

$$\mathbf{e}_{qa} = \text{Encoder}(\text{``Q: } [q] \text{ [SEP] A: } [a]\text{''}) \tag{17}$$

where Encoder() is a pre-trained sentence transformer model, specifically all-mpnet-base-v2 (Song et al., 2020). We perform approximate nearest neighbor search within hierarchical buckets organized by category, subcategory, and question type—ensuring that retrieved candidates originate from similar spatial reasoning contexts. From the top-$k$ neighbors $\mathcal{N} = \{(q_i, a_i)\}_{i=1}^{k}$, we extract their answers as initial candidate distractors $\mathcal{C}_0 = \{a_1, \ldots, a_k\}$.

To balance relevance and discriminability, we apply a band-pass similarity filter:

$$\mathcal{C}_1 = \{c \in \mathcal{C}_0 \,|\, s_{\min} \leq \text{sim}(\mathbf{e}_{a^*}, \mathbf{e}_c) \leq s_{\max}\} \tag{18}$$

where $\mathbf{e}_{a^*} = \text{Encoder}(a^*)$ and $\mathbf{e}_c = \text{Encoder}(c)$ are answer embeddings, and $\text{sim}(\cdot, \cdot)$ denotes cosine similarity. Based on preliminary experiments, we set $s_{\min} = 0.25$ and $s_{\max} = 0.92$.

**Stage 2: Multi-Criteria Validation**. We apply a cascaded filtering pipeline to ensure distractor quality:

To prevent semantic equivalence between distractors and the correct answer, we employ a fine-tuned Natural Language Inference model to compute entailment scores in both directions:

$$\text{NLI}(a^* \to c) = P(\text{entailment} \,|\, a^*, c) \tag{19}$$

We reject candidates where $\max(\text{NLI}(a^* \to c), \text{NLI}(c \to a^*)) \geq \tau_{\text{nli}}$, where $\tau_{\text{nli}} = 0.7$ in our implementation.

Each candidate is scored against the question using a BERT-based cross-encoder $R(q, c)$ to ensure contextual plausibility. We retain candidates with scores above a threshold $\tau_{\text{rel}}$.

To avoid redundancy among distractors, we remove candidates that mutually entail each other with score $\geq \tau_{\text{nli}}$. Specifically, for each candidate $c_i$, we check:

$$\forall c_j \in \mathcal{C}_{\text{selected}} : \max(\text{NLI}(c_i \to c_j), \text{NLI}(c_j \to c_i)) < \tau_{\text{nli}} \tag{20}$$

**Stage 3: Diversity-Aware Selection**

We formulate distractor selection as a Maximal Marginal Relevance optimization problem to balance relevance and diversity. Starting with an empty selected set $\mathcal{S} = \emptyset$, we iteratively select distractors via:

$$c^* = \arg \max_{c \in C \setminus S} \left[ \lambda R(q, c) - (1 - \lambda) \max_{c_j \in S} \text{sim}(e_c, e_{c_j}) \right]. \tag{21}$$

where $R(q, c)$ denotes the relevance score computed by a cross-encoder model, specifically ms-marco-MiniLM-L-6-v2 (Reimers & Gurevych, 2019), which is fine-tuned on the MS MARCO dataset for passage ranking. The parameter $\lambda \in [0, 1]$ controls the trade-off between relevance and diversity, and is set to $0.7$ based on validation experiments. This greedy selection iteratively builds a set of $k$ distractors.

**Stage 4: LLM-Based Refinement and Augmentation**

When retrieval-based methods yield insufficient candidates ($|\mathcal{C}| < k$), we employ a two-phase large language model (LLM) strategy. We prompt GPT-4o-mini with prompt 7 to paraphrase and improve the linguistic quality of existing candidates while preserving semantic content. For insufficient candidates, we generate additional distractors to meet the requirement. This generation process employs few-shot prompting, with the specific prompt template provided in Prompt 8.

**Stage 5: NLI Model Based Refinement**. To ensure the rigorous quality of the answer and distractors, we implement an automated iterative optimization pipeline designed to eliminate logical loopholes and statistical artifacts. We employ a DeBERTa-v3-large-mnli model to detect textual entailment between options, ensuring mutual exclusivity. Simultaneously, we employ a rule-based approach to identify significant length imbalances relative to the correct answer. When an issue is flagged—whether it be a logical dependency or a length cue—an LLM-based agent (e.g., GPT-4o) functions as an optimizer. It receives a targeted prompt to rewrite the specific problematic distractors based on the identified error type: prompt 9 is used for logical dependency cues, while prompt 10 is applied to length problems. This refinement process operates

recursively for a maximum of $K = 3$ rounds; if an item fails validation, the sampling temperature is incrementally increased (e.g., by $0.2$ per step) to encourage greater linguistic diversity in subsequent generations. By strictly preserving the correct answer while dynamically adjusting the distractors, this pipeline effectively minimizes the presence of trivial heuristics, resulting in a more robust and challenging evaluation benchmark.

## F.5. Quality-control Pipeline

User-centric spatial understanding fundamentally requires reasoning about visual spatial relationships rather than exploiting linguistic biases or prior knowledge. To ensure our evaluation genuinely measures models' ability to interpret dynamic spatial relationships from visual input—rather than their capacity for statistical pattern matching or commonsense reasoning alone—we implement a rigorous quality control pipeline using GPT-5.

**Blind test procedure.** We provide GPT-5 with only the question and options (without any visual information), asking it to select the correct option and provide reasoning. We use the prompt in § 11 for this process. If GPT-5 answers correctly, this indicates the distractor options are ineffective and need to be regenerated.

This quality control addresses a critical challenge in user-centric evaluation: when users ask spatial questions about their environment (e.g., "Can I reach the cup?"), they expect models to perceive and reason about the actual spatial configuration, not merely guess based on typical scenarios. Our pipeline ensures that answering our questions requires genuine visual spatial understanding, mirroring the grounded perception users need in real-world.

**Analysis of distractor failures.** Through this process, we identified two primary reasons why distractors fail: (1) *Semantic implausibility*: distractor options are obviously unreasonable and can be easily eliminated; (2) *Information imbalance*: correct answers typically align better with linguistic conventions and intuitive expectations, providing richer contextual information, whereas distractors often lack such natural qualities.

**Distractor regeneration strategy.** To address these issues, we adopt the following strategy: rather than simply generating alternative distractors that GPT-5 cannot eliminate (prompt § 12), we have GPT-5 generate two statements with semantic richness comparable to or exceeding the original correct answer. These statements are then used to replace the original distractor options, ensuring information balance across all options.

Finally, we conduct manual inspection of the dataset to verify that questions relate to dynamic spatial relationships, evidence timestamps are correct, and answers are clear. Throughout this process, we use a UI as shown in Figure 14.

## F.6. Human Performance and QA

To provide an upper-bound reference for model evaluation, we conducted a controlled human study on a subset of the benchmark. The study was designed to ensure full comparability with model tasks while capturing the challenges of temporal and spatial reasoning in video-based QA.

### F.6.1. TASK ALIGNMENT

Human evaluators were presented with **exactly the same questions** as the models, including multiple-choice format, query timestamps, and candidate answer options. The test set comprised **20 short, 20 medium, and 20 long videos**, totaling **985 questions**. All evidence annotations and ground-truth answers were removed to prevent prior knowledge from influencing responses. Participants were instructed to base their answers strictly on visual evidence rather than guesswork. The questions were drawn from four categories with roughly balanced proportions: **Trajectory & Movement (24.1%)**, **Position & Orientation (22.5%)**, **Proximity & Reachability (27.5%)**, and **Category & Quantity (25.9%)**.

### F.6.2. FULL-CONTEXT VIDEO ACCESS

In this evaluation setting, human subjects were allowed to watch the entire video or a relevant temporal window, with the freedom to pause, replay, and revisit segments as needed. This setup reflects the practical scenario where understanding requires integrating information across multiple frames, emphasizing *spatial memory and recall* rather than instantaneous perception. By providing unrestricted temporal access, the evaluation captures more accurately the reasoning processes involved in complex video comprehension tasks.

F.6.3. INDEPENDENT ANSWERING

Three human experts independently completed the full set of questions, producing responses in isolation without discussion or consultation. Each question was assigned a single final answer to maintain compatibility with model output formats. This procedure prevents mutual influence between evaluators and mitigates potential group bias, providing a more robust and objective estimate of human performance.

F.6.4. INTER-ANNOTATOR AGREEMENT

The consistency of answers between the three evaluators was carefully assessed to identify ambiguous or ill-posed questions. Items showing significant disagreement were either rephrased for clarity or excluded from the benchmark entirely. By retaining only questions with high inter-annotator agreement, the evaluation ensures both the reliability of the data and the interpretability of the results, supporting a fair and rigorous comparison with model performance.

F.6.5. HUMAN PERFORMANCE METRIC

Human performance was quantified using **average accuracy**, calculated as the mean accuracy across the three experts for all retained questions. The human evaluators achieved an **average accuracy of 91.0%**, demonstrating the **feasibility and solvability** of the dataset based on visual evidence alone. Performance across categories and subcategories is summarized in Table 3 , providing detailed reference scores for comparison with model results.

# G. Dataset Analysis

## G.1. Representative Failure Cases

We present illustrative examples for each error category, highlighting typical failure patterns and their underlying causes.

**Type I: Spatial Reasoning Failure.**   These errors reflect models' inability to construct accurate 3D spatial representations from ego-centric viewpoints. Consider the following cases:

- *Q: "Can I reach the comb now?"*
  **Model:** B (*"No"*)   **Ground Truth:** A (*"Yes"*)
  The model fails to estimate reachability, likely due to inaccurate depth perception or inability to model the ego-centric agent's physical constraints.
- *Q: "Where is the sofa relative to me?"*
  **Model:** C (*"On my right"*)   **Ground Truth:** D (*"Behind me"*)
  The model confuses spatial directions (front/back, left/right), suggesting reliance on 2D image features rather than true 3D scene understanding.

**Type II: Temporal State Tracking Failure.**   These errors indicate models' difficulty in tracking state changes and reconstructing event histories. Examples include:

- *Q: "How many times have I taken the stairs so far?"*
  **Model:** A (*"1 time"*)   **Ground Truth:** B (*"2 times"*)
  The model loses temporal context, failing to aggregate trajectory information across the video.
- *Q: "How many balls did my friend throw at the bowling machine just now?"*
  **Model:** B (*"3 balls"*)   **Ground Truth:** E (*"6 balls"*)
  The model undercounts events, suggesting it may only attend to recent frames rather than maintaining a cumulative state tracker.

**Type III: Visual Memory Amnesia.**   These errors arise when models forget visual attributes of objects seen earlier in the video:

- *Q: "What color is the trash can in the room?"*
  **Model:** B (*"Blue"*)   **Ground Truth:** D (*"Green"*)
  The model hallucinates object attributes, possibly because the trash can appeared only in early frames and its features were lost in subsequent processing.

- *Q: "Where did I just put the white cup?"*
  **Model:** D (*"On the counter"*)   **Ground Truth:** A (*"On the table"*)
  The model cannot recall the location of an object from earlier in the video, indicating degradation of visual memory over time.

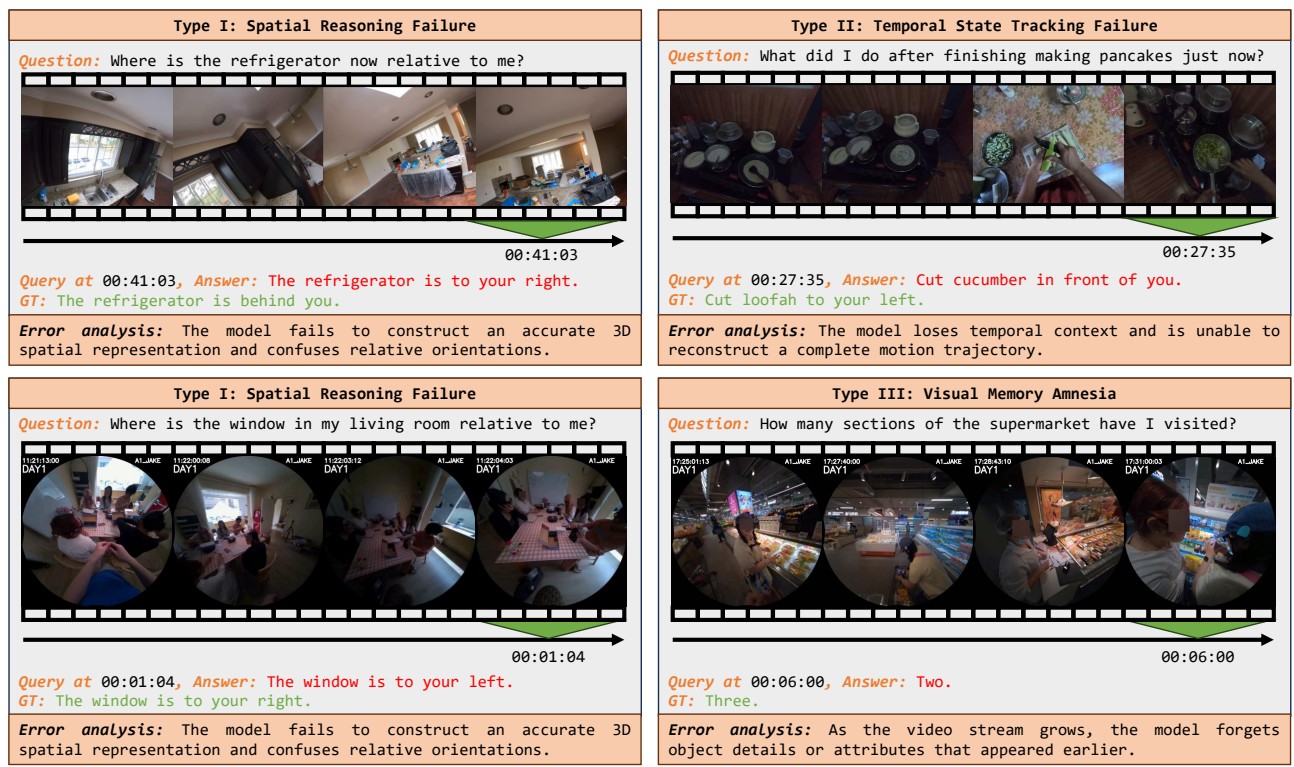

*Figure 15.* Qualitative Analysis of Typical Failure Cases. The visualization categorizes model errors into spatial reasoning failures (Type I) , temporal state tracking errors (Type II) , and visual memory amnesia (Type III). These examples highlight the current challenges in maintaining long-term spatial-temporal consistency and memory retention during extended video reasoning tasks.

### G.2. Insights and Implications

Our analysis reveals three fundamental capability gaps in current VLMs:

**1. Lack of 3D Spatial Grounding.**    The high error rate on spatial reasoning tasks (57.4%) indicates that models primarily rely on 2D image feature matching rather than explicit 3D geometric understanding. The frequent confusion of relative positions (e.g., *left* vs. *right*, *front* vs. *back*) suggests that models lack structured representations of scene geometry and ego-centric spatial relationships.

**2. State Memory Bottleneck.**    Despite improvements in context window length, models struggle to distinguish between *instantaneous observations* ("what I see now") and *accumulated states* ("what has changed over time"). The 61.2% error rate on temporal tracking questions—particularly those involving counting and trajectory reconstruction—reveals a fundamental limitation in maintaining coherent state representations across long videos.

**3. Long-Range Visual Forgetting.**    Models exhibit significant forgetting of visual details from earlier frames, especially as video length increases. This suggests that current KV-cache compression or token pruning strategies may inadvertently discard semantically critical information. The 51.2% error rate on visual memory tasks indicates that temporal recency bias dominates over semantic importance in current architectures.

**Directions for Improvement.**    Based on these findings, we propose three potential research directions:

1. **3D-Aware Pretraining:** Incorporate depth maps or camera pose as auxiliary modalities during training to encourage learning of 3D spatial relationships.
2. **Explicit State Tracking:** Design dedicated memory modules (e.g., external memory banks or state trackers) that operate independently of the LLM context window to explicitly track object quantities and state transitions.
3. **Temporal Logic Alignment:** Curate fine-tuning data containing explicit state-change chains (e.g., *object A on table → A picked up → A in hand*) to strengthen models' causal and temporal reasoning.

These directions align with broader efforts in video understanding research and suggest that addressing egocentric video challenges requires architectural innovations beyond simply scaling context length or visual resolution.

## G.3. Qualitative Analysis

Figure 16 displays representative visual scenes from our benchmark, showcasing sixteen different environment types including banks, tourist attractions, sidewalks, stairs, subway stations, warehouses, campuses, laboratories, spacious roads, workshops, wilderness areas, convenience stores, roadways, grocery stores, pathways, and kitchens. This figure demonstrates the breadth of naturally occurring everyday scenes captured in our dataset, spanning both indoor and outdoor environments. The environmental diversity highlights our dataset's ecological validity for evaluating spatial reasoning and memory in contexts that reflect common daily-life interactions and navigation scenarios.

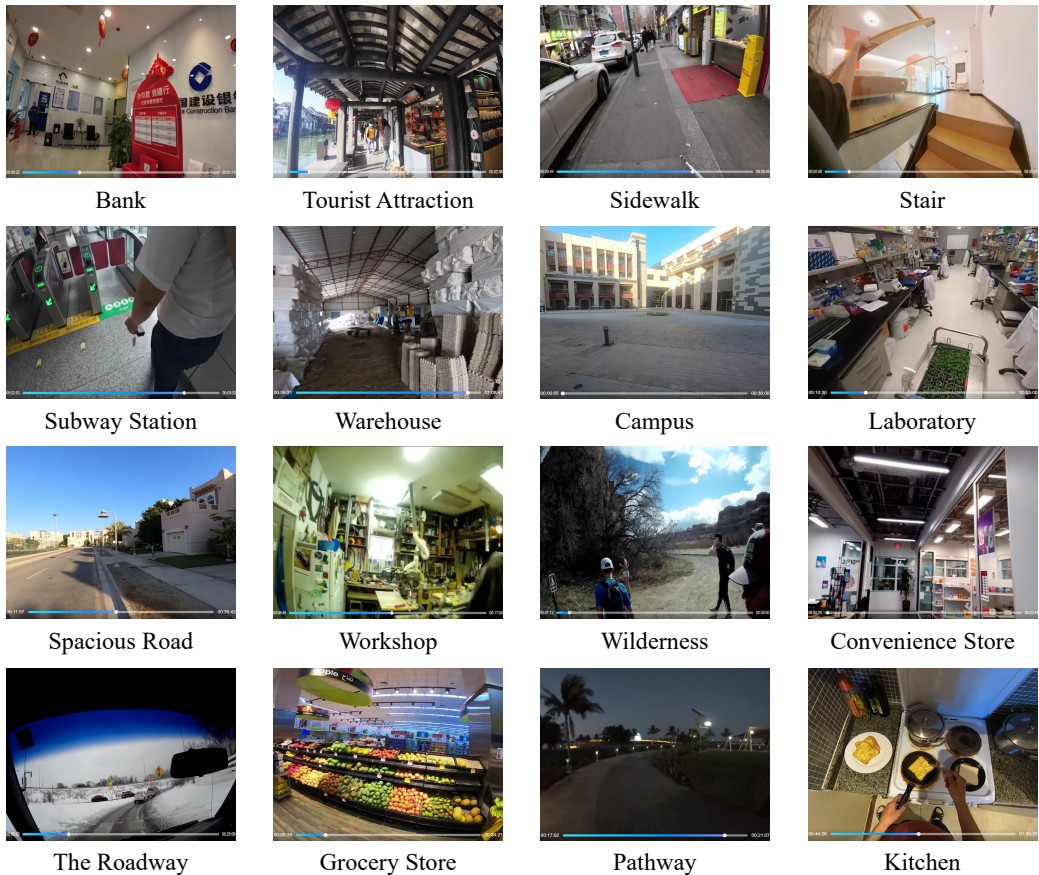

*Figure 16.* Representative visual scenes. We showcase representative examples from our benchmark to illustrate the diversity of naturally occurring everyday scenes, reflecting common daily-life environments and interactions that are relevant for evaluating spatial reasoning and memory.

Table 7 presents the eight tasks supported in our benchmark, including example questions, answer options, and ground truth labels for each task. This table demonstrates the diversity of spatial reasoning and memory tasks covered by our benchmark, ranging from egocentric position estimation to quantity change tracking. The comprehensive task taxonomy highlights our

*Table 7.* Example questions and answers (QA) for the 8 tasks supported in our benchmark.

| Task | Example questions | Example answer options | GT |
|---|---|---|---|
| Ego-centric Position | Where is the kitchen now relative to me? | ['A. It's now behind you.' 'B. It's now to your left.' 'C. It's now in front of you.' 'D. It's now on your right.' 'E. It's now below you.'] | "D" |
| Relative Position & Orientation Memory | Where was it before I picked up your phone for the first time? | ['A. The phone was on the shelf on the second floor, to the left of the television remote.' 'B. The phone was in the drawer beside the computer on the first floor, next to a notebook.' 'C. The phone was on the ground on the first floor, in front of the remote control instead.' 'D. The phone was on the table on the first floor, to the left of the coffee mug.' 'E. The phone was on the ground on the first floor, to the right of the remote control.'] | "E" |
| My Trajectory | What was my path after buying ice cream? | ['A. Go straight to the left.' 'B. Go straight to the right.' 'C. Go straight ahead to the store.' 'D. Go straight to the left, then walk to the park.' 'E. Go straight to the left, then walk to the café.'] | "A" |
| Object Trajectory | What is the trajectory of the foam board from the moment it is picked up until it is set down? | ['A. After being lifted from the sofa, it moved in a straight line to the table right in the dining room.' 'B. After being lifted from the cushion, it moved in a straight line to the floor right in the center of the living room.' 'C. After being picked up from the rug, it moved in a straight line to the shelf right in the hallway.' 'D. After being lifted from the floor, it moved in a straight line to the chair right in the bedroom.' 'E. After being picked up from the countertop, it moved in a straight line to the cabinet right in the kitchen.'] | "B" |
| Distance Comparison Memory | Which is closer to me, the mirror or the dining table? | ['A. Mirror.' 'B. Dining table.'] | "A" |
| Reachability Judgment | Can I reach the sink now? | ['A. You cannot reach the sink now.' 'B. You can reach the sink now.'] | "B" |
| Object Category Recall | What vegetables on the table did I take to wash? | ['A. You took the cucumbers on the table to wash.' 'B. You took the carrots on the table to wash.' 'C. You took the loofah on the table to wash.' 'D. You took the radishes on the table to wash.' 'E. You took the peppers on the table to wash.' ] | "C" |
| Quantity Change Tracking | How many pieces of dough are still on the kitchen table now? | ['A. There are three pieces of dough left on the kitchen table in front of me because I left them there.' 'B. There is one piece of dough left on the kitchen table to my right because I moved the rest to the oven.' 'C. There is no dough left on the kitchen table near me because I put it in the refrigerator.' 'D. There are two pieces of dough left on the kitchen table in front of me because I forgot to put any in the refrigerator.' 'E. There are five pieces of dough left on the kitchen table to my left because I placed them on the countertop instead.'] | "D" |

dataset's ability to evaluate multiple aspects of embodied spatial intelligence through natural language question-answering formats.

Figure 17 illustrates representative examples of streaming memory annotations for object recall and quantity tracking tasks, showing video frame sequences with timestamped evidence and corresponding question-answer pairs. This figure demonstrates how our benchmark evaluates temporal reasoning by requiring models to track object states and quantities across different timestamps within continuous video streams. The evidence-based annotation structure highlights our dataset's emphasis on grounding spatial reasoning in specific visual observations over time.

Figure 18 shows qualitative results of streaming spatial reasoning in egocentric video for the "My Trajectory" task, with examples from both indoor grocery store and outdoor urban environments. This figure illustrates cases where the model performs continuous spatial transformations as the video stream unfolds. The diverse environmental contexts highlight our dataset's coverage of real-world navigation scenarios across varied indoor and outdoor settings.

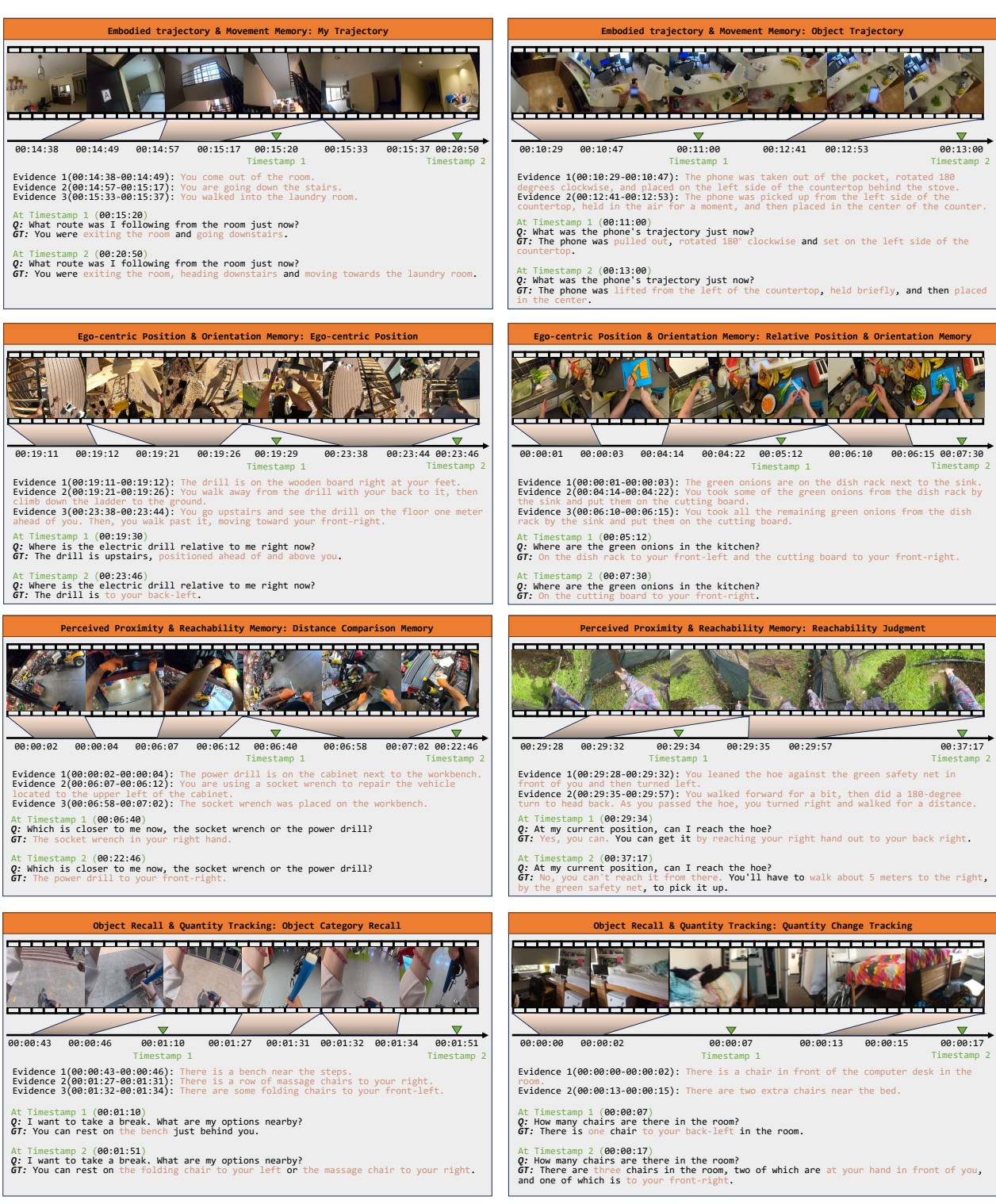

*Figure 17.* This figure illustrates representative examples of streaming memory annotations across four major categories and eight subcategories, specifically showcasing indoor and outdoor scenarios. It details evidence-based ground truth for tasks such as Object Trajectory and Egocentric Position across multiple timestamps.

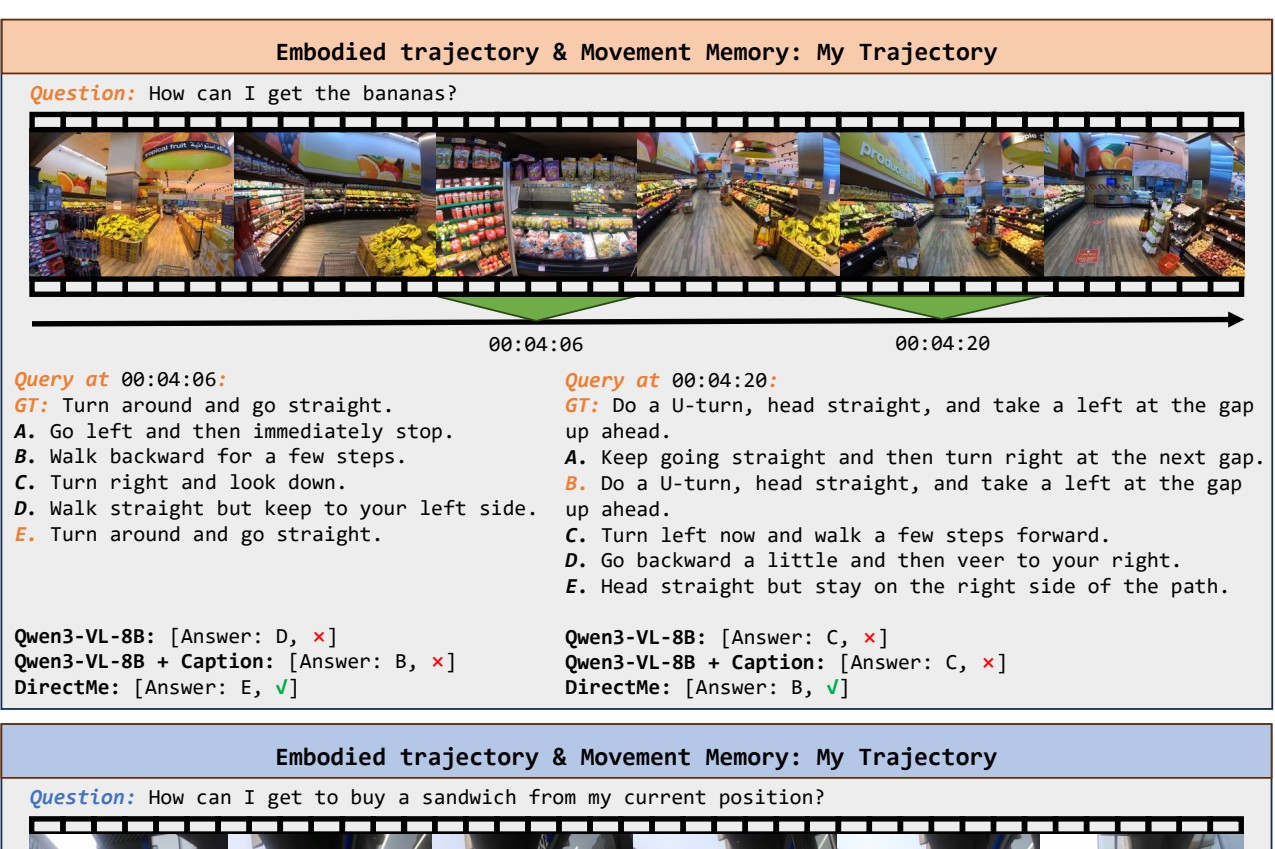

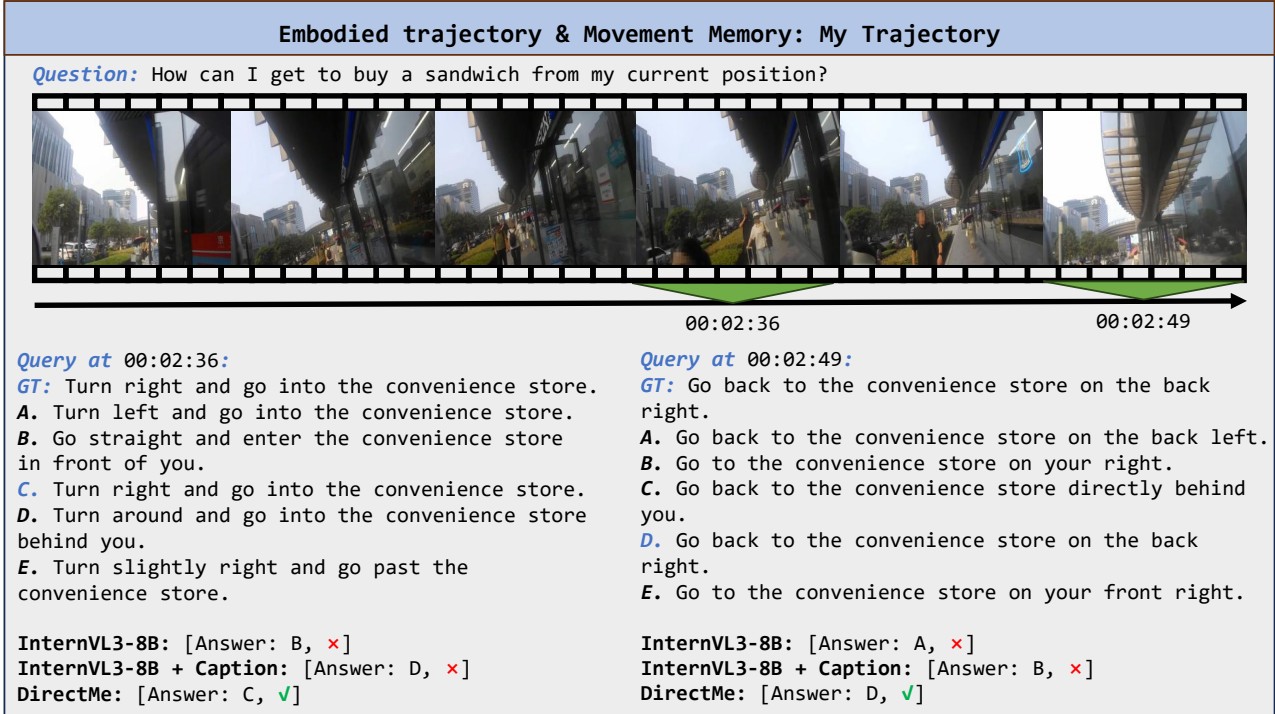

*Figure 18.* Qualitative results of **streaming spatial reasoning in egocentric video**. This figure illustrates our model's capability to perform continuous spatial transformations as the video stream unfolds. By leveraging **Movement Memory**, the system dynamically updates the agent's spatial orientation relative to the environment across various timestamps.

