# OpenReview forum: "Keep It in Mind: User Centric Continual Spatial Intelligence Reasoning in Egocentric Video Streams"
_ICML.cc/2026/Conference — ICML 2026 regular_

### Official Review · Reviewer_tbdc · 2026-03-10

**Soundness:** 2
**Presentation:** 2
**Significance:** 2
**Originality:** 2
**Overall Recommendation:** 4
**Confidence:** 2

**Summary:**

The paper introduce UCS-Bench, an novel dataset for testing User-centric Continual Spatial intelligence in egocentric video streams. To improve performance on this task, the paper further proposes DirectMe, a memory-based framework designed to support the tracking and recall of object locations.

**Compliance With Llm Reviewing Policy:**

Affirmed.

**Key Questions For Authors:**

See weakness.

**Limitations:**

yes

**Strengths And Weaknesses:**

**Strengths:**

The benchmark construction pipeline is well designed and appears to be carefully organized. The motivation behind DirectMe is also intuitive: maintaining an explicit user-centric spatial memory seems like a reasonable direction for handling long-horizon egocentric spatial reasoning.

**Weaknesses**

The fairness of the experimental settings is questionable. In Table 3, most baseline models are evaluated with 64 input frames, while DirectMe is evaluated under a 1 fps setting. Although the paper may intend this to reflect a streaming setup, the comparison is still not fully clean.

The ablation study is still limited. The paper only reports an ablation on Desc. and Graph memory, which is helpful but not sufficient to clarify the contribution of the major components in the full pipeline. DirectMe involves several important stages, including depth estimation, camera pose estimation , multi-object tracking, and scene-graph construction, but their individual effects are not disentangled. It remains unclear which part is actually driving the performance gains, and whether all of these modules are necessary.

The paper does not report any computation cost analysis for DirectMe. Given that the method relies on multiple integrated models and processing stages, an efficiency analysis would be important for evaluating its practicality .

---

> ### Author Rebuttal · Authors · 2026-03-31
>
> **We greatly appreciate your time and constructive suggestions. Below, we provide point-by-point responses and will further incorporate the relevant analyses and experiments in the final version.**
>
> #### **W4.1:** "The fairness of Table 3 is questionable ... baselines use 64 frames while DirectMe uses 1 fps."
>
> **R4.1:** We understand this concern and wish to clarify that our comparisons are fair in two matched settings:
>
> First, for models evaluated under a uniform 64-frame sampling setting (Table 3), we also evaluate DirectMe with the same 64-frame input, so the comparison is made under the same frame budget:
> | Streaming Model |Frame| Overall Acc.(%) |
> | ------- | --------- | --------------: |
> | DirectMe  (w/InternVL-3) | 64 | 52.7|
> | DirectMe  (w/Qwen3-VL) | 64 |  53.5 |
>
> Second, for streaming models, DirectMe is compared under the same 1fps setting, as also shown in R1.2, where attaching DirectMe to representative streaming backbones (Dispider, Videochat-online) yields consistent gains.
>
> Together, these results show that the improvement of DirectMe does not come from a favorable input protocol, but from the structured scene-memory design itself. Finally, we wish to note that different models often have their own optimal number of input frames (depending on their training protocols), and our investigations show that 64 is the best for most models.
>
> #### **W4.2:** "The ablation study is still limited. The paper only reports an ablation on Desc. and Graph memory, which is helpful but not sufficient to clarify the contribution of the major components in the full pipeline. DirectMe involves several important stages, including depth estimation, camera pose estimation, multi-object tracking, and scene-graph construction, but their individual effects are not disentangled. It remains unclear which part is actually driving the performance gains, and whether all of these modules are necessary."
>
> **R4.2:**  Thanks for this suggestion. In addition to the original Desc./Graph ablation, we further disentangle the major components of DirectMe by ablating key stages in the full pipeline:
> | Variant | Overall Acc. (%) |
> |---|---:|
> | Full DirectMe | 54.1 |
> | w/o scene graph construction| 45.4|
> | w/o depth | 51.2|
> |w/o pose | 49.5|
> | w/o object tracking | 48.3|
>
> The results show that the gains are not driven by any single module: removing any major component causes a clear drop. Among them, the largest drops come from removing the scene graph, pose, and object tracking, indicating that the main benefit comes from an evolving pose-anchored scene memory with consistent cross-time object identity. Depth also contributes, though less strongly. Overall, DirectMe’s improvement comes from the full structured memory pipeline, rather than any isolated stage.
>
> #### **W4.3:** "The paper does not report any computation cost analysis for DirectMe. Given that the method relies on multiple integrated models and processing stages, an efficiency analysis would be important for evaluating its practicality."
>
> **R4.3:** Thanks for the suggestion. We have added a detailed efficiency analysis in **v8RJ：W1.1**. Although DirectMe involves multiple integrated models and processing stages, it follows an offline question-agnostic incremental memory update + online QA design. As reported in  **v8RJ：W1.1**, each 10-second window requires 3.3 s for offline scene-graph memory update, while online QA takes only 0.9–2.4s/query on a single A100, depending on the chosen MLLMs. Importantly, the offline memory update is query-independent. Therefore, user-perceived latency depends solely on online QA reasoning rather than on the cost of offline updates. We will include a detailed analysis in the final version.

---

> > ### Author Rebuttal · Reviewer_tbdc · 2026-04-05
> >
> > My concerns have been adequately addressed.

---

> > > ### Author Response · Authors · 2026-04-06
> > >
> > > We sincerely thank Reviewer **tbdc** for the positive score and the thoughtful feedback. We really appreciate your careful reading and your recognition of the novelty and technical value of our work. Your comments were both encouraging and helpful, prompting us to better clarify the paper's motivation, evaluation, and broader significance.

---

### Official Review · Reviewer_enn8 · 2026-03-10

**Soundness:** 3
**Presentation:** 4
**Significance:** 2
**Originality:** 3
**Overall Recommendation:** 4
**Confidence:** 4

**Summary:**

The paper presents two main contributions: (i) a VQA dataset (UCS-Bench) for evaluating the spatial reasoning capabilities of LLMs when applied to egocentric videos, and (ii) a training-free framework (DirectMe) that builds a dynamic and spatially grounded scene graph from the video input to support spatial reasoning in LLMs.

The benchmark focuses on very long range spatial and temporal reasoning and combines different reasoning categories that were treated separately by previous benchmarks. These QA categories cover different aspects, such as reasoning about the relative position and viewpoint of the user, trajectories, reachability of the objects in the scene, objects persistence and counting. Data collection is supported by human annotators and a filtering pipeline based on natural language reasoning and blind tests using GPT5.

For each question, DirectMe retrieves the most relevant sub-graph from the full scene graph, which is then fed to the LLM along with a descriptive caption of the video. When tested with Qwen3-VL and InternVL3, DirectMe provides significant improvements, demonstrating the effectiveness of the structured scene graph based representation.

**Compliance With Llm Reviewing Policy:**

Affirmed.

**Final Justification:**

The rebuttal and discussion clarified my doubts about evidence grounding and human evaluation. The motivation for introducing a new benchmark for user-centric spatial understanding are sound and the benefits of the graph-based approach are well demonstrated on downstream QA benchmarks.
Therefore, i will raise my score accordingly.

**Key Questions For Authors:**

1. The subgraph retrieval approach allows to identify the portion of the scene graph most relevant to the given query. Is there any way to evaluate the actual relevance of the retrieved subgraph?

2. Could DirectMe be adapted to a true online setting (as discussed in the strengths and weaknesses paragraph) by building the scene graph representation online?

3. Can you provide more details about the subgraph retrieval approach?

**Limitations:**

yes

**Strengths And Weaknesses:**

UCS-Bench is not the first benchmark to evaluate spatial understanding in LLMs, even when considering the egocentric setting. As presented in Table 2, existing benchmarks already consider most reasoning perspectives, even though UCS-Bench unifies all of them in a more comprehensive benchmark. In this regard, it is not clear whether the introduction of a new benchmark is fully motivated. Furthermore, DirectMe is only evaluated on the newly introduced UCS-Bench. Evaluating the effectiveness of the proposed framework on well-established terms would provide a more sound evaluation of the method.

Lines 360-364 seem to contradict one of the main contributions, i.e., the introduction of the DirectMe framework, by stating that “strong general visual-language reasoning and instruction following capabilities can be more critical than specialized architectural designs for generalized spatial modeling”. Indeed, the graph representation proposed in DirectMe is a specialized architectural designed for spatial reasoning.
The scene graph provides a structured representation of the spatial relations of the entities in the scene and the QA categories in UCS-Bench attempt to capture different reasoning capabilities. However, the model does not explicitly track the generated answers down to its supporting evidence (i.e., objects and relations supporting the answer).

The claim that the QA pairs follow an online streaming setting is questionable, since the scene graph representation is built offline from the entire video and only queried at inference time. As an example, this definition differs from the approach used in OST-Bench, in which the model is asked to reason over incrementally received observations. The subgraph retrieval mechanism is not very clearly described.

The analysis of the failure cases is limited. The structured representation offered by DirectMe could support a more fine-grained evaluation of the different error types.

---

> ### Author Rebuttal · Authors · 2026-03-31
>
> **We appreciate your valuable time and comments. We address the major concerns below and will add related clarifications and implementation details in the revision.**
>
> #### **W3.1:** "UCS-Bench is not the first spatial benchmark .... DirectMe is only evaluated on UCS-Bench ... evaluation on established benchmarks is needed."
>
> **R3.1:** We wish to clarify that we do not claim UCS-Bench as the first benchmark for spatial understanding. Rather, we emphasise a novel **user-centric, continual** setting for streaming ego vision, distinct from existing spatial understanding benchmarks. The new challenge focuses on modeling object locations (often out-of-sight at the question time) relative to a user's real-time movements under ego-motion (L39-L44, L135-L140, and Sec 3.3). Our benchmark is the first to challenge such a setting in the community. Moreover, while we list several featured dimesions for better comparison with existing works in Table 2, the specific meaning may not be identical. For example, our 'dynamic' highlight ego-motion of the camera wearers, beyond only the visible moving objects in the videos as defined in most existing benchmarks. We will clarify such distinctions in the revision.
>
> We further evaluate DirectMe on the established benchmark **VSI-Bench**, assuming all questions are asked at the end of each video. The results show consistent gains over the baselines.
> | Model | Acc. |
> |---|---:|
> | InternVL3 |42.1|
> | InternVL3 w/DirectMe |47.0|
> | QWEN3-VL |58.1|
> | QWEN3-VL w/DirectMe|61.5|
>
> #### **W3.2:** "Lines 360–364 seem to contradict one of the main contributions ... no explicit answer-to-evidence grounding."
> **R3.2:** The cited sentence summarizes our observations about **existing** general-purpose and specialized models in **generalizing** to the new user-centric continual spatial modeling challenge in UCS-Bench, i.e, generalized spatial modelling. Our point is **not** that specialized spatial design is unnecessary, but that existing spatial models are not tailored to the **user-centric continual** setting targeted by UCS-Bench. We will rephrase the sentence for better clarity. Finally, our method actually enjoys the benefits of both general-purpose and specilized designs.
>
> Regarding evidence grounding, 10 volunteers reviewed retrieved subgraphs for 100 randomly sampled QA pairs and found that about 82% of them contain the spatial evidence needed to answer. We will expand this study and add visualizations in the revision.
>
> #### **W3.3:** "The `online streaming' claim is questionable ... scene graph is built offline ... retrieval mechanism is unclear."
>
> **R3.3:**  Thanks. Online streaming QA differs from offline video QA, where the entire video is assumed available and the answer remains the same for repeated questions. In our online streaming setting, each question can only access the video contents up to the question moment for answer. The scene graph process is question-independent and thus can be done at the background (offline when no question is triggered). We do not build a static scene graph for the entire video, but incrementally maintain and update the scene graph at each timestamp (a fixed window). That is, a question at time *t* will only access the graph constructed up to time *t*.
>
> For subgraph retrieval, we first parse the question into a structured query containing the target entity, relation type, and temporal scope. We then ground the target entity to candidate object nodes in the scene graph up to time 𝑡. If multiple candidates exist, we disambiguate them using the recency of the last confident observation and the persistence of the associated track, which helps identify the most reliable node as the anchor. Based on this anchor, we retrieve a compact pose-aware subgraph with the current ego pose and relevant object-object / object-place relations, which is then provided to the MLLM for answer generation. More implementation details will be added in the revision.
>
> #### **W3.4:** "Failure-case analysis is limited ... the structured representation could support finer-grained error analysis."
> **R3.4:**  Thank you for this suggestion.  We manually inspect 100 failed QA cases and assign each case to its primary failure mode. Since module-level ground-truth attribution is unavailable, we report the dominant error type per failure case.
>
> |Failure source |Ratio (%) |
> |---| ---: |
> |Retrieval failure |16|
> |Tracking failure |38|
> |Ego-pose relation failure |25|
> |Reasoning failure on correct evidence |21|
>
> These results suggest that the main challenge is maintaining user-centric spatial state, rather than retrieval alone. Most failures come from tracking (38%) and ego-pose relations (25%), indicating that temporally consistent memory and viewpoint alignment are the hardest parts. By comparison, retrieval failure (16%) is less common, while reasoning failure on correct evidence (21%) indicates that final spatial inference remains nontrivial even when relevant evidence has been retrieved.

---

> > ### Author Rebuttal · Reviewer_enn8 · 2026-04-03
> >
> > I thank the authors for their rebuttal which partially clarified my doubts.
> >
> > I appreciate the human-based evaluation of the retrieved subgraphs and the error type analysis. However, this evaluation remains limited to a small number of samples. Given the scope and the objectives of the proposed benchmark, a more systematic evaluation would be very valuable for understanding the limitations of the current models and how/where a structured representation can help. The rebuttal also provides little detail regarding how human evaluation was conducted. Is there a more measurable approach to verify evidence grounding?

---

> > > ### Author Response · Authors · 2026-04-06
> > >
> > > We deeply appreciate the reviewer’s active engagement and valuable feedback.
> > >
> > > **1. Expanded Human Evaluation Details & Error Analysis**
> > > To further address the concern about the limited sample size, we significantly expanded the human study from 100 to 500 failed QA cases.
> > >
> > > We trained 10 university students to review the QAs, the corresponding time-stamped video clips, and retrieved subgraph texts, and diagnose the root-causes for the failure answers. For instance, if an initial tracking error subsequently leads to an incorrect ego-pose calculation, the case is exclusively classified as "Tracking failure". The updated results based on this root-cause analysis are as follows:
> > >
> > > |Primary failure source| Ratio (%) |
> > > |:-|:-:|
> > > |Tracking failure|40|
> > > |Ego-pose relation failure|23|
> > > |Retrieval failure|20|
> > > |Reasoning failure on correct evidence|17|
> > >
> > > These results again suggest that the main challenge is maintaining the user-centric spatial state, rather than retrieval alone. Most failures stem from tracking (40%) and ego-pose relations (23%), indicating that temporally consistent memory and viewpoint alignment are the key bottlenecks. By comparison, retrieval failure (20%) is less common, while reasoning failure on correct evidence (17%) suggests that these remaining errors arise not from missing spatial priors, but from the model’s final answer selection stage. Likely causes include over-reliance on salient visual cues in the key frames and imperfect alignment between the textual prior and the candidate options.
> > >
> > >
> > > **2. Systematic Evaluation via ConceptGraphs Paradigm**
> > > Because our dynamically evolving spatial memory is inherently open-ended, there is no single, deterministic ground truth for the intermediate graph. We therefore adopt an extrinsic evaluation protocol, in the same spirit as prior open-ended structured representation work such as **ConceptGraphs** [1], i.e., we evaluate the representation by how well it supports downstream reasoning rather than by exact graph matching. To make this evaluation scalable beyond small human studies, we further adopt an **LLM-as-a-Judge protocol**, which is well-suited to open-ended structured outputs.
> > >
> > > To explicitly show **how and where** the structured representation helps, we use two measurable metrics:
> > >
> > > **Retrieved Evidence Grounding Score.** For each question in UCS-Bench, we provide GPT-4o with the question, the ground-truth answer, and the **retrieved subgraph**, and ask it to score how well this retrieved subgraph grounds the **target entity** and the **spatial relation** using a fixed 5-level scale ($S_i \in \{0, 0.25, 0.5, 0.75, 1.0\}$), ranging from missing or contradictory evidence (Score: 0.0) to exact grounding (Score: 1.0). We then average the scores over all evaluated questions:
> > >
> > > $\text{Grounding Score} = \frac{1}{N}\sum_{i=1}^{N} S_i \times 100\%$,
> > >
> > > where (*N*) is the total number of evaluated questions.
> > >
> > > **Downstream QA Accuracy.** The final QA accuracy of the downstream MLLM using the retrieved representation on UCS-Bench. Here, **Base** denotes the same backbone MLLM (**Qwen3-VL-8B**) without structured scene-graph retrieval, while **Ours** denotes the same backbone augmented with our retrieved structured representation (DirectMe).
> > >
> > > |Task Category|Retrieved Target Entity Grounding Score(%)|Retrieved Spatial Relation Grounding Score(%)|Downstream QA Acc.(%) (Base/Ours)|
> > > |:-:|:-:|:-:|:-:|
> > > |Pos & Ori|80.4|69.7|41.2/50.2|
> > > |Traj.& Mov|78.8|71.5|45.5/56.8|
> > > |Prox & Reach|82.6|74.9|51.8/62.8|
> > > |Rec & Qua|75.3|66.4|43.7/50.5|
> > > |Overall|78.7|70.0|45.0/54.1|
> > >
> > > As shown above, the **retrieved target-entity grounding scores** are consistently higher than the **retrieved spatial-relation grounding scores** across all task categories.
> > > This is expected: identifying the relevant entity is only the first step, while correctly grounding its **relation to the user** is more demanding, since it depends on whether the scene graph captures spatial structure, temporal updates, and ego-pose alignment. Therefore, **spatial-relation grounding** is the more challenging and more diagnostic metric.
> > >
> > > Meanwhile, the consistent improvements in **downstream QA accuracy** suggest that this grounded structured evidence is useful for end-task reasoning. Notably, the strong results on **Proximity & Reachability** show that the retrieved scene graph preserves user-centric spatial relations well, while gains on **Position & Orientation** and **Trajectory & Movement** highlight its value for more dynamic reasoning involving temporal updates and ego-pose alignment.
> > >
> > > [1] Gu, Qiao, et al. "Conceptgraphs: Open-vocabulary 3d scene graphs for perception and planning." 2024 IEEE International Conference on Robotics and Automation (ICRA). IEEE, 2024.

---

### Official Review · Reviewer_V28Z · 2026-03-12

**Soundness:** 3
**Presentation:** 3
**Significance:** 3
**Originality:** 3
**Overall Recommendation:** 4
**Confidence:** 3

**Summary:**

This paper aims to address the lack of user-centric continual spatial reasoning in egocentric video streams. Motivated by the fact that humans naturally track object locations relative to their own moving perspective over long time spans, the paper argues that current visual AI systems struggle severely with this dynamic spatial memory, especially when objects leave the field of view.

The mian contribution is a new benchmark, UCS-Bench, containing over 170 hours of egocentric video and 7.4K manually annotated QA pairs that explicitly test spatial memory over time. To tackle this benchmark, the authors propose DirectMe, a training-free framework that extracts depth, pose, and object detections to construct a persistent, structured spatial memory.  During inference, relevant subgraphs and keyframes are retrieved from this memory and passed to Multimodal LLMs to answer spatial queries. Evaluations are conducted across 18 MLLMs.

**Compliance With Llm Reviewing Policy:**

Affirmed.

**Key Questions For Authors:**

- How do we envision this system being adapted for true real-time, streaming applications? Are there specific computational bottlenecks preventing online, frame-by-frame execution?

- Does the scene graph memory actually solve the core temporal state-tracking issue, or is it just providing slightly better static spatial priors to the MLLM?

**Limitations:**

Yes

**Strengths And Weaknesses:**

### Strengths

- Overall, the proposed benchmark, UCS-Bench, is rigorous, featuring repeated questions over time, long-form videos, and blind-choice bias control to ensure models cannot rely on linguistic priors.

- Additionally, the paper provides excellent and insightful error analysis, accurately identifying visual memory amnesia and temporal tracking failures as the core bottlenecks of current models.

### Weaknesses

 - DirectMe seems to relies heavily on the offline preprocessing components, including depth estimation, camera pose tracking, and object detection. It is likely that errors in any of these early stages will propagate directly into the spatial memory graph, corrupting the reasoning process.

---

> ### Author Rebuttal · Authors · 2026-03-31
>
> **Thanks a lot for your valuable time and constructive feedback, we address the concerns point-by-point below and will add related analyses and experiments in the final version.**
>
> #### **W2.1:** "DirectMe seems to relies heavily on the offline preprocessing components, including depth estimation, camera pose tracking, and object detection. It is likely that errors in any of these early stages will propagate directly into the spatial memory graph, corrupting the reasoning process."
>
> **R2.1:** Thanks for the insightful point. Errors from upstream modules (e.g., depth estimation, camera pose tracking, and object detection) can, in principle, propagate into the scene graph. However, our results suggest that such error accumulation is limited in practice and does not overturn the main benefit of DirectMe.
> This is because DirectMe mainly reasons over user-centric relational spatial states (e.g., left/right, front/behind, reachable/not reachable), rather than requiring highly precise 3D reconstruction at every frame. Moreover, many UCS-Bench questions depend on temporally accumulated evidence across multiple observations, which makes the final reasoning less sensitive to isolated upstream errors in individual frames.
>
> To directly test this robustness, we replace DepthAnything3 with a weaker depth estimator (DepthAnything) and Grounding-DINO with a weaker detector (YOLO-World-S), while keeping the rest of the pipeline unchanged. The resulting drops are modest:
>
> | Setting                    | Acc. (%)    |
> | -------------------------- | ----------- |
> | DirectMe (DA3+G-DINO)     | 54.1  |
> | w/DepthAnything (DA)          | 52.9 (-1.2) |
> | w/YOLO-World-S (YOLOW) | 52.1 (-2.0) |
> | w/(DA+YOLOW)  | 51.4 (-2.7) |
>
> Importantly, even with weaker upstream modules, DirectMe still clearly outperforms the baseline without structured memory, indicating that upstream noise is manageable rather than dominant. Therefore, the current upstream modules are sufficient to support user-centric continual spatial reasoning in our framework.
>
> #### **W2.2:** "How do we envision this system being adapted for true real-time, streaming applications? Are there specific computational bottlenecks preventing online, frame-by-frame execution?"
>
> **R2.2:**  We have added a detailed system analysis of real-time deployment in **v8RJ: W1.1**. As quantified in **R1.1**, the question-independent background memory update in DirectMe operates faster than the incoming stream, indicating that maintaining the structured spatial memory is not the primary computational bottleneck. Instead, the dominant source of user-perceived latency is query-time MLLM inference, which takes 0.9-2.4s per query on a single A100 GPU, depending on the chosen backbone.
>
> More importantly, DirectMe is not designed to run full MLLM reasoning on every frame. Rather, it incrementally updates memory in the background as new observations arrive and invokes the QA model only when a user query is issued, which is a more practical design for streaming assistant scenarios. Therefore, rather than claiming a fully real-time end-to-end system, we view the current results as showing that structured memory maintenance is already efficient enough for streaming settings, while the remaining bottleneck for frame-level responsiveness lies mainly in final answer generation. Further reductions would come from lighter QA backbones or stronger hardware.
>
> #### **W2.3:** "Does the scene graph memory actually solve the core temporal state-tracking issue, or is it just providing slightly better static spatial priors to the MLLM?"
>
> **R2.3:**  Thanks for this suggestion. **Table 4** suggests that DirectMe does more than provide better static spatial priors; it substantially improves the core temporal-state-tracking ability (**Traj. & Mov.**). Static per-frame description augmentation yields only marginal gains, and can even hurt performance in some cases (InternVL: 45.8→47.1; Qwen3-VL: 45.5→44.2), whereas explicit scene-graph memory brings much larger improvements (54.5 / 56.8). This shows that the gain in temporal state tracking comes primarily from the evolving scene-graph memory, rather than from slightly better static spatial priors alone.

---

> > ### Author Rebuttal · Reviewer_V28Z · 2026-04-04
> >
> > Thanks the authors for the response.

---

> > > ### Author Response · Authors · 2026-04-06
> > >
> > > We sincerely thank Reviewer **V28Z** for the positive score and the supportive feedback. We truly appreciate the time and care you put into reading the paper, as well as your recognition of its novelty and technical contribution. Thank you again for your recognition and valuable comments.

---

### Official Review · Reviewer_v8RJ · 2026-03-13

**Soundness:** 2
**Presentation:** 2
**Significance:** 3
**Originality:** 3
**Overall Recommendation:** 4
**Confidence:** 5

**Summary:**

This paper explores User-centric Continual Spatial Intelligence in egocentric video streams. Recognizing that previous video MLLM research has primarily focused on static, offline, or third-person perspectives, the authors introduce UCS-Bench to evaluate a model's ability to incrementally integrate streaming first-person observations and maintain an evolving egocentric spatial reference. To enhance this capability, they propose the DirectMe framework, which constructs a structured spatial memory from egocentric streams. Experimental results demonstrate that DirectMe significantly improves performance on UCS-Bench, although a substantial gap between state-of-the-art models and human-level performance remains.

**Compliance With Llm Reviewing Policy:**

Affirmed.

**Final Justification:**

Thank the authors for the further analysis. Now I'm fully aware that user-centric continual spatial intelligence is significant. I will raise the score accordingly.

**Key Questions For Authors:**

Refer to weaknesses.

**Limitations:**

Refer to weaknesses.

**Strengths And Weaknesses:**

Strengths

-	The paper introduces UCS-Bench, a novel benchmark that effectively bridges a critical research gap in user-centric continual spatial intelligence within egocentric video streams.
-	Experimental results reveal that existing video MLLMs struggle with user-centric continual spatial reasoning. This finding underscores the significance of the proposed research and its underlying motivation.

Weaknesses

-	Lack of real-time validation: The authors propose DirectMe as a framework targeting real-time egocentric video streams. However, experimental results lack this "real-time streaming" validation. As the framework relies on an offline multi-modal processing pipeline involving computationally expensive external models, it is important to include any quantitative analysis regarding inference latency, throughput, or the cumulative overhead of updating the scene graph.
-	Lack of validation for general video stream understanding capabilities: The UCS-Bench remains primarily diagnostic, focusing on low-level spatial queries such as position, orientation, and reachability. Although user-centric continual spatial intelligence seems important, the significance of such ability is not fully validated against high-level practical tasks found in established online egocentric benchmarks like Ego4D, StreamingBench, or OVO Bench. Specifically, a robust spatial memory should inherently improve performance by providing stable spatial anchors, yet no comparison is provided against existing streaming models on these standard metrics.

---

> ### Author Rebuttal · Authors · 2026-03-31
>
> **Lots of thanks for your valuable time and thoughtful comments. We carefully address all concerns below and will incorporate related analyses/clarifications in the revision:**
>
> #### **W1.1:** '' Lack of real-time validation, ..., experimental results lack this "real-time streaming" validation. ... important to include any quantitative analysis regarding inference latency, throughput, or the cumulative overhead of updating the scene graph.''
>
> **R1.1:** Thank you for highlighting this. As described in Sec. 4.2, the “offline” part of DirectMe is in fact an asynchronous, incremental visual memory update: the incoming video stream is processed in fixed-time windows, and the structured memory is continuously maintained independent of questions. This stage is done offline prior to the question. For online QA, whenever a query is triggered, DirectMe only retrieves a subgraph and a small set of supporting keyframes to generate the answer, rather than reprocessing the full video stream for each question.
> | Module | Latency |
> | :--- | :--- |
> | Offline Memory Update (per 10s video window) | 3.3 s |
> | ├─ DepthAnything3 | 0.6 s |
> | ├─ Multi-object Tracking | 1.2 s |
> | ├─ Scene Describer | 1.3 s |
> | └─ Scene Graph Construction | 0.2 s |
> | Online QA Latency (per question) | 0.9–2.4 s |
>
> As shown above, on a single A100 GPU, processing a 10-second video clip requires 3.3s for cumulative overhead memory updates, which is well below the input duration and therefore enables real-time throughput. The processing time can be further reduced to 1.5s with a parallel implementation of the video multimodal processing modules.
> For online QA, latency is mainly determined by the MLLM inference stage: it is 2.4s/query with the current model (Qwen3-VL-8B), and 0.9s/query with a faster model (VideoChat-Online). Therefore, similar to other streaming video-LLMs, online MLLM inference is the bottleneck for real-time interaction, rather than offline memory update. We will clarify this in revision.
>
> #### **W1.2:**  ''Lack of validation for general video stream understanding capability, ..., the significance of such ability is not fully validated against high-level practical tasks found in established online egocentric benchmarks .... A robust spatial memory should..., yet no comparison is provided against existing streaming models on these standard metrics.''
>
> **R1.2:** Thank you for the comment. While the suggested benchmarks are significant for advancing **general-purpose** streaming video understanding, UCS-Bench pursues an **orthogonal goal** and targets a specific capability: **user-centric continual spatial intelligence** in a streaming setting. All questions in UCS-Bench are practical (e.g., assisting the visually impaired) and align well with recent trends in understanding video spatial intelligence toward embodied assistance. Compared with existing spatial QA and streaming QA tasks (see Table 2), UCS-Bench is the first to study the capability to locate objects (previously observed in the camera stream but often out of view at the moment of the query) relative to the user's real-time movements.
>
> Meanwhile, our observations in Sec. 4.3 suggests that solving this task requires higher-level spatial reasoning over temporally accumulated observations together with the user’s current egocentric state. This is exactly the ability UCS-Bench is designed to evaluate, and it is also the key motivation for introducing a structured spatial memory. In this sense, our benchmark is not intended to  **replace** established online egocentric benchmarks, but rather to  **complement** them by targeting a capability dimension that is currently underexplored on these streaming benchmarks. Therefore, evaluation on those benchmarks would provide only partial, less direct evidence for our main claim, since they do not explicitly measure whether a model can maintain and update user-centric spatial memory for previously seen but currently out-of-view objects.
>
> To verify the robustness of our spatial memory, we additionally integrate it into representative streaming models to evaluate them on UCS-Bench: **Dispider** and **VideoChat-online**. The results below show that the structured memory brings clear gains, demonstrating that it provides a robust spatial reference for existing streaming models.
> We will also revise the paper to clarify this scope more explicitly.
>
> | Model | Baseline | + DirectMe | Gain |
> |---|---:|---:|---:|
> |Dispider |39.1|46.7|+7.6|
> | VideoChat-online |37.5|45.8|+8.3|

---

> > ### Author Rebuttal · Reviewer_v8RJ · 2026-04-03
> >
> > I appreciate the authors' response. However, I still remain unconvinced regarding the significance of the proposed user-centric continual spatial intelligence.
> >
> > I am aware that the target ability or intelligence is novel and unaddressed in previous works and that the proposed UCS-Bench is specifically designed for it. However, I question whether this specific intelligence itself is impactful enough to warrant its own benchmark. I believe the suggested intelligence is not orthogonal to general streaming scenarios while the benchmark is. To justify the necessity of exploring it, it must be demonstrated that this user-centric continual spatial intelligence is not just a niche capability but is fundamentally significant across more general scenarios. However, this research only deals with the UCS-Bench in evaluation.
> >
> > Thus, I maintain the score as it is.

---

> > > ### Author Response · Authors · 2026-04-06
> > >
> > > We gratefully thank the reviewer for recognizing the novelty of our research. To further clarify its broader impact, we wish to highlight that our work aligns closely with recent trends in **video spatial intelligence** for embodied and wearable AI assistants (e.g., VSI-Bench, VSI-Super).
> > > Existing streaming benchmarks such as StreamingBench and OvOBench are largely **non-egocentric**. Their major focus is **real-time video semantic parsing** for live shows/games, surveillance videos, etc., not adhering to the ego-view embodied scenario. User-centric continuous spatial (UCS) intelligence is not a niche capability; it is practically important for AR/VR, robotics, autonomous systems, and personal assistants.
> > >
> > > We contribute both the UCS-Bench dataset and the DirectMe solution, a dedicated framework for measuring and advancing this UCS intelligence, providing a necessary foundation for progress in next-generation embodied and companion-style AI systems. While our solution is evaluated on UCS-Bench, we mainly highlight the **Continual Spatial Scene Graph** module as a plug-and-play design over a general-purpose ego streaming QA framework to enhance continual spatial reasoning, which, in principle, does not affect general QA capability. We validated this through an extended experiment and comparison on the general-purpose ego-streaming QA benchmark **VStream-QA (VS-Ego)** [1]. The results below show that our method also surpasses previous SOTA on the general-purpose ego streaming QA benchmark.
> > >
> > > |Model|Acc (VS-Ego[ICCV'25])| Acc (UCS-Bench)
> > > |:-|:-:|:-:|
> > > |Flash-VStream[ICCV'25]|59.0|28.6|
> > > |DirectMe|63.2|54.1|
> > >
> > > Therefore, we view UCS-Bench not as a benchmark for a niche setting, but as a necessary complement to existing streaming evaluations, targeting an important egocentric dimension that existing benchmarks largely underexplored.
> > >
> > > [1] Zhang, Haoji, et al. "Flash-vstream: Memory-based real-time understanding for long video streams." ICCV, 2025.

---

### Decision · Program_Chairs · 2026-04-30

**Decision:**

Accept (regular)

**Comment:**

The paper introduces UCS-Bench, a user-centric continual spatial reasoning benchmark in long egocentric video streams, and DirectMe, a training-free spatial memory framework that substantially improves multimodal LLM performance on this task. All reviewers lean positive (4 x weak accepts). After rebuttal, reviewers confirm that their main concerns are fully addressed. The rebuttal and follow-up comments provide additional experiments that strengthen the case that the benchmark fills a gap in egocentric spatial intelligence and that DirectMe is a generally useful plug-in spatial memory module. Given the unanimous consensus, the recommendation is to accept.